# ACS-20/FATP4 mediates the anti-ageing effect of dietary restriction in *C. elegans*

Zi Wang[1], Lina Zou[1], Yiyan Zhang [1,2], Mengnan Zhu[3], Shuxian Zhang[3], Di Wu[4], Jianfeng Lan[5], Xiao Zang[1,2], Qi Wang[1,2], Hanxin Zhang[1], Zixing Wu[1], Huanhu Zhu [3] & Di Chen [1,2,6] ✉

Dietary restriction is an effective anti-ageing intervention across species. However, the molecular mechanisms from the metabolic aspects of view are still underexplored. Here we show ACS-20 as a key mediator of dietary restriction on healthy ageing from a genetic screen of the *C. elegans* acyl-CoA synthetase family. ACS-20 functions in the epidermis during development to regulate dietary restriction-induced longevity. Functional transcriptomics studies reveal that elevated expression of PTR-8/Patched is responsible for the proteostasis and lifespan defects of *acs-20*. Furthermore, the conserved NHR-23 nuclear receptor serves as a transcriptional repressor of *ptr-8* and a key regulator of dietary restriction-induced longevity. Mechanistically, a specific region in the *ptr-8* promoter plays a key role in mediating the transcription regulation and lifespan extension under dietary restriction. Altogether, these findings identify a highly conserved lipid metabolism enzyme as a key mediator of dietary restriction-induced lifespan and healthspan extension and reveal the downstream transcriptional regulation mechanisms.

Numerous studies have demonstrated that ageing can be modulated by highly conserved genes and pathways, and the molecular mechanisms involve the interaction with environmental factors such as nutrients[1]. Besides the genetic modulation of ageing, environmental changes can also have a significant impact on ageing. Dietary restriction (DR), reduced food intake without malnutrition, is one of the most robust environmental interventions that not only extend lifespan but also improve healthspan[2–4]. Originally reported in a study that rats with reduced food intake had significantly increased longevity[5], DR has been shown to prolong the longevity of *S. cerevisiae*[6,7], *C. elegans*[8], *Drosophila*[9], as well as *M. mulatta*[10]. Although there has not been direct evidence to connect DR with human longevity, many studies showed the beneficial effects of DR in human health[4]. Thus, it is of high significance to investigate the molecular mechanisms of DR in model

animals for common downstream targets, which can be subject to pharmaceutical interventions to delay ageing.

Genetic studies using model animals have helped to reveal the molecular mechanisms of DR. Several DR methods have been developed using *C. elegans* as a model[2,3]. Hypomorphic mutants of *eat-2*, which encodes a subunit of the nicotinic acetylcholine receptor, have reduced food in-take thus serving as a genetic mimic of DR[8]. DR can be achieved by directly manipulating the bacterial food, such as feeding animals with diluted food either in liquid culture[11,12] or on solid agar plates[13,14], complete food deprivation[15], and intermittent fasting[16]. Using these DR regimens, researchers have identified multiple mediators of DR-induced longevity, including transcription factors SKN-1/ Nrf2[12], PHA-4/FOXA[11], DAF-16/FOXO[13], HSF-1/HSF[15], and NHR-62/ HNF4α[17], stress regulators EGL-9/PHD[14] and IRE-1/IRE[14], energy

[1]Model Animal Research Center of Medical School, MOE Key Laboratory of Model Animals for Disease Study, Nanjing University, Nanjing, Jiangsu 210061, China. [2]Zhejiang University-University of Edinburgh Institute, Zhejiang University School of Medicine, Haining 314400, China. [3]School of Life Science and Technology, ShanghaiTech University, Shanghai 201210, China. [4]Institute of Drug Discovery and Development, Center for Drug Safety Evaluation and Research, Zhengzhou University, Zhengzhou, Henan 450001, China. [5]Affiliated Hospital of Guilin Medical University, Guilin, Guangxi 541001, China. [6]Department of Colorectal Surgery, The Second Affiliated Hospital, Zhejiang University School of Medicine, Hangzhou, Zhejiang 310058, China. ✉ e-mail: dic@intl.zju.edu.cn

homeostasis regulator AAK-2/AMPKα[13] and so on. Mutations in these regulators either completely or largely block DR-induced longevity. The downstream mechanisms of DR involve mitochondrial functions, various stress responses, autophagy, proteostasis, and metabolism[11–13,15,16,18,19].

Although lipid metabolism plays important roles in ageing and health, it has not been clear whether specific enzymes from the complex lipid metabolic network are required for the anti-ageing effect of DR. The acyl-CoA synthases (ACSs) family of enzymes function in the initial step of fatty acid metabolism by catalyzing the synthesis of fatty acyl-CoAs, which serve as metabolic intermediates[20] to enter subsequent lipid metabolic processes, such as lipid synthesis and degradation, lipid modification of proteins, and generation of transcriptional regulators and signaling molecules. In *C. elegans*, there are 21 *acs* genes (*acs*−1–7, 9-22), many of which have not been well characterized for their roles in lipid metabolism or other biological processes.

In this study, we set out to examine whether the *acs* genes are required for the DR-induced lifespan extension by an RNAi-based genetic screen. We found that ACS-20 is required for the lifespan and healthspan extension by DR. Spatiotemporal analysis reveals that ACS-20 functions in the epidermis during development to regulate DR-induced longevity. Functional genomics studies demonstrate that the *acs-20* mutation leads to ER stress under DR. *ptr-8*, which encodes an ortholog of Patched from the Hedgehog pathway, is transcriptionally upregulated in the *acs-20* mutant, and it is responsible for the lifespan and proteostasis defects of the *acs-20* mutant under DR. Studies on the transcriptional regulation show that the NHR-23/RORA nuclear hormone receptor serves as not only a transcriptional repressor of *ptr-8*, but also a key mediator of DR-induced longevity. Moreover, a specific region of the *ptr-8* promoter has been mapped as an important *cis*-regulatory element for the NHR-23-mediated transcriptional regulation of *ptr-8*. Deletion of this region alleviates the DR-induced longevity defect when animals are subject to *acs-20* or *nhr-23* knockdown. Therefore, ACS-20 functions through NHR-23 to restrict the expression of PTR-8 to ensure the beneficial effects of DR on lifespan and healthspan.

## Results

### ACS-20 is a key mediator of DR-induced lifespan and healthspan extension

To characterize the roles of lipid metabolism in the anti-ageing effect of DR, we performed an RNAi-based genetic screen of 21 acyl-CoA synthetase (*acs*) genes for their effects on lifespan under DR conditions in *C. elegans*. Compared to the control RNAi, the *acs-20* RNAi treatment most significantly suppresses the prolonged longevity by DR (Supplementary Fig. 1). *acs-20* encodes the *C. elegans* ortholog of the fatty acid transporter 4 (FATP4), mutations in which leads to abnormal lipid metabolism in the epidermis and disease such as ichthyosis prematurity syndrome (IPS) in mammals[21,22]. In *C. elegans*, the *acs-20* mutant shows abnormalities in cuticle structures[23].

To further validate the role of ACS-20 in DR-induced lifespan extension, wild-type N2 and *acs-20* deletion mutant animals were treated with bacterial food at different concentrations ($1.0 \times 10^8$–$1.0 \times 10^{12}$ cfu/ml) from day 1 adulthood for survival assays. Unlike N2 animals, which exhibit more lifespan extension with food dilution until the optimal DR condition ($1.0 \times 10^9$ cfu/ml), the *acs-20* mutant shows no significant changes in lifespan under different nutrient conditions (Fig. 1a). The *eat-2* mutant, which has reduced feeding and significantly extended lifespan, has been widely used as a genetic mimic of DR[8]. The *acs-20* mutant completely abolishes the prolonged longevity of *eat-2* animals (Fig. 1b). Therefore, ACS-20 is a key mediator of DR-induced longevity.

In addition to lifespan, healthspan is also an important assessment of healthy ageing[1]. To explore the role of *acs-20* in healthspan

regulation, we first examined thermotolerance phenotypes of N2, *eat-2*, *acs-20*, and *eat-2; acs-20* mutants by heat shock at 35 °C for 10 h. The *eat-2* mutant shows significantly increased resistance to the heat stress compared to the wild-type N2, whereas the *acs-20* mutation completely blocks the protective effect of *eat-2* (Fig. 1c). Polyglutamine (polyQ) expansion-induced proteotoxicity has been linked with several degenerative diseases[24]. We took advantage of an established polyQ model, in which a transgene expresses polyQ (Q35) fused with YFP in the body wall muscle[25], to test the effect of *acs-20* on proteostasis. Compared to the wild type, the *eat-2* mutant shows significantly less Q35::YFP aggregates and delayed paralysis due to the proteotoxicity in body wall muscle cells (Fig. 1d–f). However, the protective effect of *eat-2* is abrogated by the *acs-20* mutation (Fig. 1d–f). Taken together, these results demonstrate that ACS-20 is required for DR-induced healthspan extension.

To further examine whether ACS-20 is regulated by nutrients, we performed CRISPR/Cas9-based genome editing experiment to knock in GFP::3 × FLAG coding sequence to the 3′ end of *acs-20*. We then compared ACS-20 protein levels between wild-type and *eat-2* mutant animals via immunoblots. The ACS-20::GFP::3 × FLAG protein levels are mildly but significantly elevated under DR (Fig. 1g, h), suggesting ACS-20 is involved in the response to low nutrient conditions.

### Temporospatial requirement of ACS-20 in DR-induced longevity

The *acs-20::gfp::3 x flag* knockin (KI) strain allows us to characterize the endogenous expression patterns of ACS-20. Confocal microscopy images show that ACS-20 is mainly expressed in the epidermal and muscular tissues (Fig. 2a), with the expression levels significantly diminished during adulthood (Fig. 2b). RT-qPCR measurement of the endogenous *acs-20* mRNA levels confirms that *acs-20* expression levels in adulthood are significantly lower than those in the third (L3) or fourth (L4) larval stages (Fig. 2c). Thus, *acs-20* is expressed in a temporospatial manner.

To determine when and where ACS-20 functions to regulate lifespan under DR, we performed RNAi knockdown experiments to examine the effects of *acs-20* reduced function on longevity. Consistent with the temporal expression patterns, the whole life *acs-20* RNAi treatment completely abolishes DR-induced lifespan extension (Fig. 2d), whereas *acs-20* RNAi only during adulthood has little effect on lifespan (Fig. 2e). Suppression of DR-induced longevity can also be achieved by the epidermis-specific *acs-20* RNAi treatment (Fig. 2f), but not by muscle-specific *acs-20* RNAi (Fig. 2g). To validate these findings, a single-copy transgene of *acs-20* driven by its own promoter was constructed for rescue experiments. The DR-induced lifespan extension deficiency of *acs-20* is fully restored by this transgene (Supplementary Fig. 2). Another single-copy transgene of *acs-20* driven by the *dpy-7* promoter, which is specifically active in the epidermis only during development[26], also completely restored DR-induced longevity in the *acs-20* mutant background (Fig. 2h). Collectively, ACS-20 functions in the epidermis during developmental stages to regulate DR-induced longevity.

### Effects of *acs-20* on lipid metabolism

ACS-20 is orthologous to the mammalian FATP4, which serves predominantly as an acyl-CoA synthetase rather than a fatty acid transporter[27]. Mutations in humans or mice FATP4 lead to lipid metabolic defects in the skin[21,22]. To characterize whether the *acs-20* mutant affects lipid accumulation, we performed Oil Red O (ORO) staining to measure neutral lipid levels in the intestine, the major metabolic and lipid storage tissue in *C. elegans*. Imaging and quantification results demonstrate that the *acs-20* mutation does not affect the intestinal lipid levels compared to the wild-type N2, whereas the *eat-2; acs-20* mutant has slightly increased lipid accumulation compared to the *eat-2* mutant (Supplementary Fig. 3a, b).

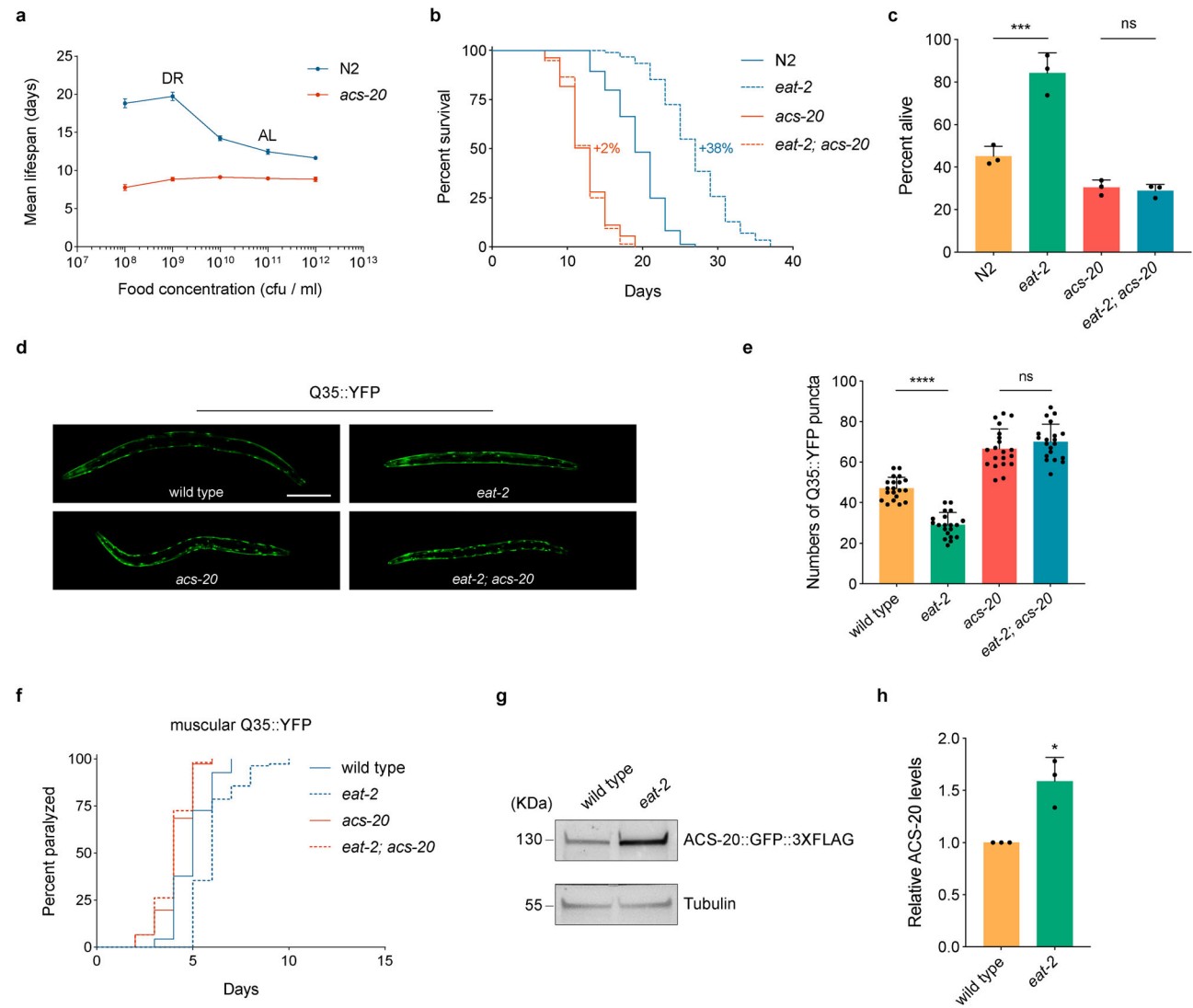

**Fig. 1 | ACS-20 is required for DR-induced lifespan and healthspan extension.**
**a** Mean lifespan of the wild-type N2 and *acs-20* mutant fed with bacterial food at different concentrations ($1.0 \times 10^8$–$1.0 \times 10^{12}$ cfu/ml). Data are represented as mean ± SD based on three independent experiments. AL, *ad libitum* ($1.0 \times 10^{11}$ cfu/ml), DR, optimal dietary restriction condition ($1.0 \times 10^9$ cfu/ml). **b** Survival curves of N2, *eat-2*, *acs-20*, and *eat-2; acs-20* mutant animals. Percentages indicate changes in mean lifespan induced by the *eat-2* mutation. N2 vs. *eat-2*, $p < 0.0001$; *acs-20* vs. *eat-2; acs-20*, $p = 0.8271$ (log-rank tests). **c** Survival percentages of N2, *eat-2*, *acs-20*, and *eat-2; acs-20* mutant animals incubated at 35 °C for 10 h. Data are represented as mean ± SD ($n = 3$). ***$p < 0.001$; ns, $p = 0.9875$ (One-way ANOVA with Turkey's multiple comparisons test). **d**, **e** Representative photographs of 3-day-old adult wild-type, *eat-2*, *acs-20* and *eat-2; acs-20* mutant animals expressing the muscular Q35::YFP (**d**) and quantification of the Q35::YFP punctae (**e**). Data are represented as mean ± SD ($n = 20$). ****$p < 0.0001$; ns, $p = 0.5231$ (One-way ANOVA with Turkey's multiple comparisons test). Scale bar, 100 μm. **f** Paralysis of wild-type, *eat-2*, *acs-20* and *eat-2; acs-20* mutant animals expressing the muscular Q35::YFP. Wild-type vs. *eat-2*, $p < 0.0001$; *acs-20* vs. *eat-2; acs-20*, $p = 0.2364$ (log-rank tests). **g**, **h** Immunoblots (**g**) and quantification (**h**) of ACS-20::GFP::3×FLAG and tubulin protein levels in the wild type and *eat-2* mutant. The ratio of band intensity of ACS-20::GFP::3×FLAG to tubulin was normalized to the wild type. Data are represented as mean ± SD ($n = 3$). *$p = 0.0111$ (two-tailed *t*-test). Source data are provided as a Source Data file.

Since ACS-20 functions in the epidermis, another lipid storage tissue in *C. elegans*, to regulate lifespan, it is important to examine whether the *acs-20* mutation affects lipid accumulation in the epidermis. However, the ORO staining method with fixed animals does not allow quantitative assessment of epidermal lipid droplets. Previous studies have shown that DGAT-2 is a membrane protein of lipid droplets, and a single-copy *gfp::dgat-2* transgene driven by the intestine-specific *vha-6* promoter can be used for quantitative measurement of intestinal lipid droplets[28]. We then constructed a single-copy *gfp::dgat-2* transgene driven by the epidermis-specific *col-12* promoter. Imaging and quantification assays show that the *acs-20* mutation does not affect epidermal lipid droplet levels either in the wild type or in the *eat-2* mutant background (Supplementary Fig. 3c, d). Altogether, these results suggest that ACS-20 might not regulate DR-induced lifespan extension via affecting lipid accumulation in metabolic tissues.

## Functional transcriptomics analysis of the *acs-20* mutant
To gain mechanistic insights of ACS-20 for its role in ageing, we conducted transcriptional profiling of the wild-type N2, *acs-20*, *eat-2*, and *eat-2; acs-20* mutant animals via mRNA-Seq. Bioinformatic analysis reveals that with the cutoff at fold changes >2 and adjusted $p < 0.01$, there are 61 upregulated and 45 down-regulated genes in *acs-20* compared to N2, whereas there are 138 upregulated and 38 down-regulated genes in *eat-2; acs-20* compared to *eat-2* (Fig. 3a and Supplementary Data 1). The 95 genes that are significantly upregulated in *eat-2; acs-20* compared to *eat-2* but show no significant changes in

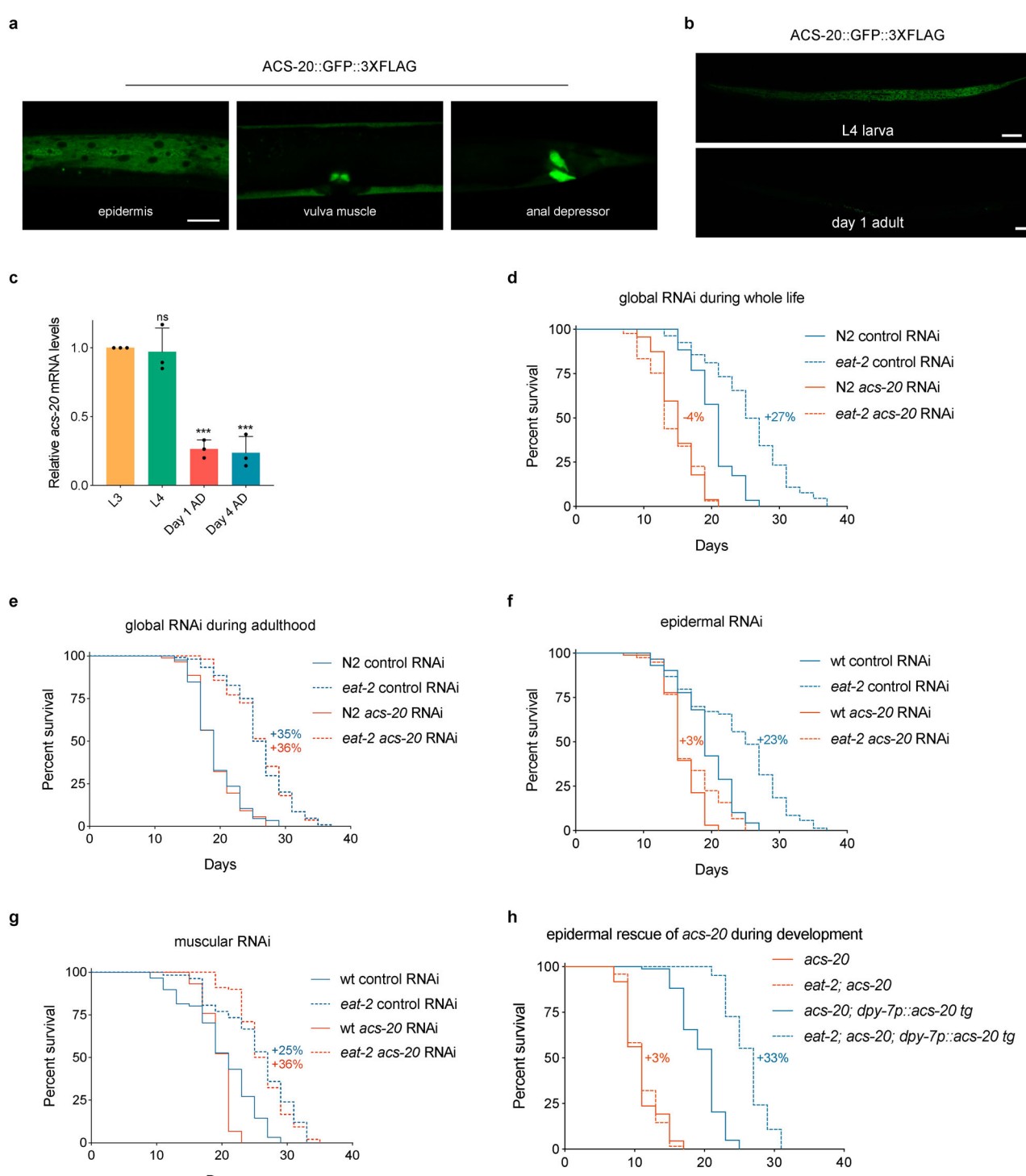

**Fig. 2 | ACS-20 functions in the epidermis during development to regulate DR-induced lifespan extension. a** Representative photographs of the endogenous ACS-20::GFP::3×FLAG expression in different tissues. Scale bar, 20 μm. **b** Representative photographs of ACS-20::GFP expression in the fourth larval stage (L4) and day 1 adulthood. Scale bar, 50 μm. **c** RT-qPCR quantification of *acs-20* mRNAs at different stages. Data are represented as mean ± SD ($n = 3$). ns, $p = 0.9880$; ***$p < 0.001$ (One-way ANOVA with Turkey's multiple comparisons test). **d**, **e** Survival curves of N2 and *eat-2* mutant animals treated with the global control or *acs-20* RNAi in whole life (**d**) or adulthood (**e**). N2 vs. *eat-2* with the control RNAi treatment, $p < 0.0001$ (whole life, adulthood); N2 vs. *eat-2* with the *acs-20* RNAi treatment, $p = 0.3093$ (whole life), $p < 0.0001$ (adulthood) (log-rank tests). **f**, **g** Survival curves of wild type (wt) and *eat-2* mutant animals treated with the epidermal (**f**) or muscular (**g**) control or *acs-20* RNAi. wt vs. *eat-2* with the control RNAi treatment, $p < 0.0001$ (epidermal RNAi, muscular RNAi); wt vs. *eat-2* with the *acs-20* RNAi treatment, $p = 0.0662$ (epidermal RNAi), $p < 0.0001$ (muscular RNAi) (log-rank tests). **h** Survival curves of *acs-20* and *eat-2; acs-20* mutants with or without the *acs-20* transgene driven by the epidermis-specific *dpy-7* promoter. *acs-20* vs. *eat-2; acs-20*, $p = 0.9773$ (without the transgene), $p < 0.0001$ (with the transgene) (log-rank tests). Percentages indicate changes in mean lifespan induced by the *eat-2* mutation. Source data are provided as a Source Data file.

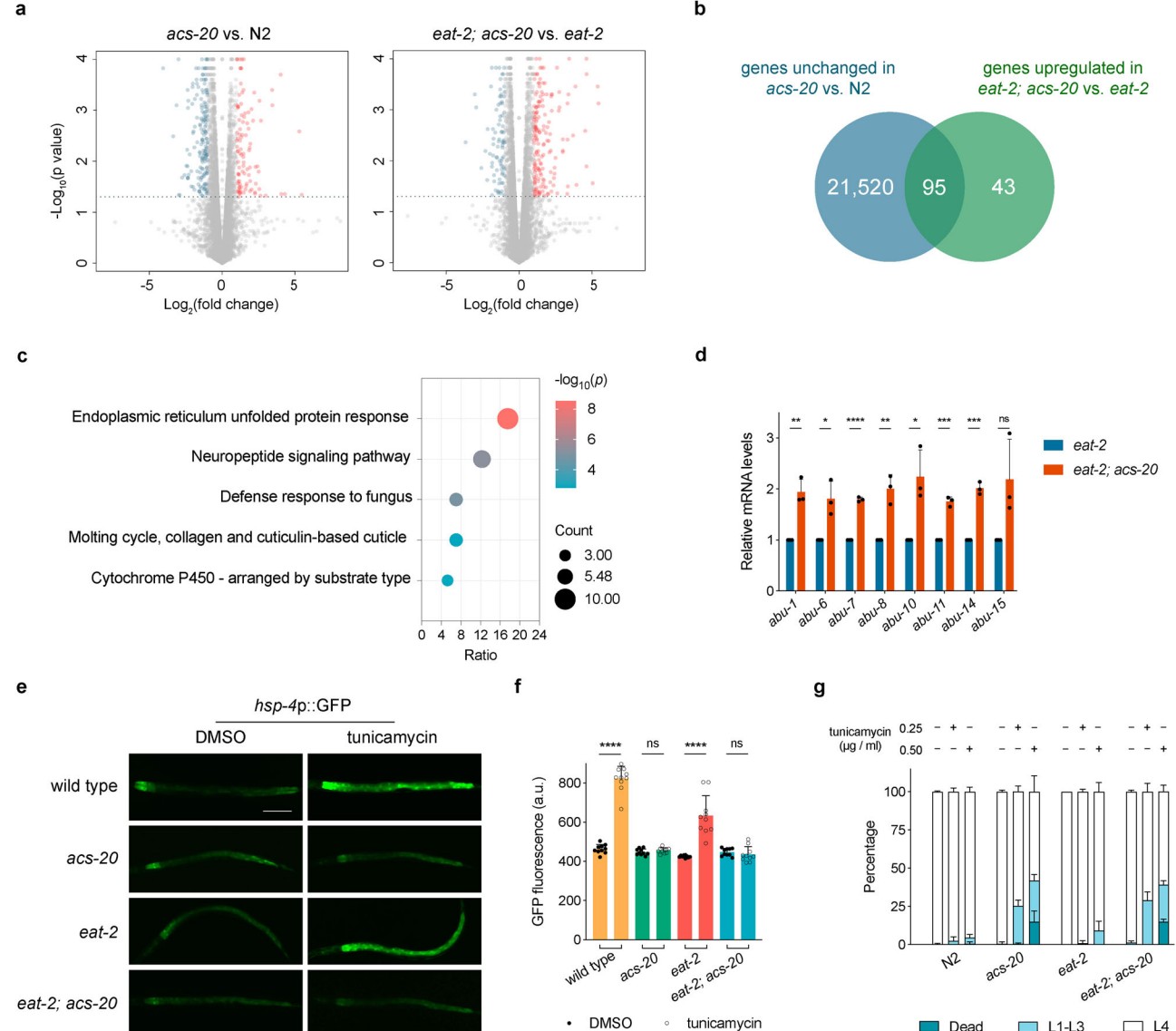

**Fig. 3 | The *acs-20* mutation causes ER stress under DR. a** Volcano plots of differentially expressed genes in the *acs-20* vs. N2 and *eat-2; acs-20* vs. *eat-2* data sets. Red and blue points represent genes with significantly increased or decreased expression, respectively ($p < 0.01$, Wald tests with Bonferroni corrections). **b** A Venn diagram comparing genes upregulated in *eat-2; acs-20* vs. *eat-2* and genes not differentially expressed in *acs-20* vs. N2. **c** Significantly enriched Gene Ontology (GO) terms (hypergeometric tests with Benjamini-Hochberg corrections) for genes significantly upregulated in the *eat-2; acs-20* mutant compared to the *eat-2* mutant. **d** RT-qPCR quantification of *abu* genes in *eat-2* and *eat-2; acs-20* mutant animals. Data are represented as mean ± SD ($n = 3$). ****$p < 0.0001$; ***$p < 0.001$; **$p < 0.01$; *$p < 0.05$; ns, $p = 0.0598$ (two-tailed *t*-tests). **e**, **f** Representative photographs of the

ER stress reporter *hsp-4p::gfp* expression in wild-type, *acs-20*, *eat-2*, and *eat-2; acs-20* animals treated with DMSO or tunicamycin (**e**) and quantification of the GFP fluorescence intensities (a.u. arbitrary unites). Data are represented as mean ± SD ($n = 10$). ****$p < 0.0001$; ns, *acs-20* $p > 0.9999$, *eat-2; acs-20* $p = 0.9993$ (One-way ANOVA with Turkey's multiple comparisons test). Scale bar, 200 μm. **g** Percentages of N2, *acs-20*, *eat-2*, and *eat-2; acs-20* mutant animals that died (dead), developmentally arrested (L1-L3) or completed development (L4) upon 0, 0.25 or 0.50 μg/ml tunicamycin treatments. Data are represented as mean ± SD ($n = 3$). For animals that completed development upon tunicamycin treatment, N2 vs. *acs-20*, $p < 0.01$; *eat-2* vs. *eat-2; acs-20*, $p < 0.0001$ (Two-way ANOVA with Turkey's multiple comparisons test). Source data are provided as a Source Data file.

*acs-20* compared to N2 were used for Gene Ontology (GO) analysis as these genes are more likely to be involved in the DR-specific response (Fig. 3b). The most significantly enriched GO term is endoplasmic reticulum (ER) unfolded protein response (Fig. 3c).

RT-qPCR experiments confirmed the mRNA-Seq results that mRNA levels of multiple ER stress-related *abu* genes are significantly elevated in the *eat-2; acs-20* double mutant compared to the *eat-2* mutant (Fig. 3d). *abu* genes encode endomembrane proteins, the expression of which is induced in ER-stressed animals when the IRE-1–XBP-1 ER unfolded protein response (ER^UPR) pathway is inactivated[29]. Tunicamycin is a protein N-glycosylation inhibitor which induces ER stress[30]. The tunicamycin treatment leads to robust activation of the

*hsp-4::gfp* reporter, which has been widely used to monitor the ER^UPR activation in *C. elegans*. Intriguingly, tunicamycin cannot activate *hsp-4::gfp* expression when *acs-20* is mutated either under normal feeding or DR conditions (Fig. 3e, f), suggesting the *acs-20* mutant might have reduced capability to deal with ER stress. Consistently, tunicamycin treatments cause more severe developmental delay and lethality when *acs-20* is deleted in the wild-type or *eat-2* mutant background (Fig. 3g). Dithiothreitol (DTT) and thapsigargin are also ER stressors that function via affecting the ER lumen oxidative environment[31] and causing ER $Ca^{2+}$ depletion[32], respectively. DTT or thapsigargin treatment leads to more severe developmental defects in the *acs-20* and *eat-2; acs-20* mutants compared to the wild-type N2 and *eat-2* mutant, respectively

(Supplementary Fig. 4). These results suggest that the *acs-20* mutation might compromise proteostasis thus increasing the sensitivity to ER stress, which contributes to the lifespan and healthspan defects under DR.

### PTR-8/Patched as a key mediator of *acs-20* for its roles in longevity and proteostasis regulation under DR

It is plausible that genes upregulated in the *eat-2; acs-20* double mutant are responsible for the lifespan and proteostasis defects of *acs-20* under DR. Therefore, we performed an RNAi screen in the *eat-2; acs-20* double mutant background to individually knock down the 95 genes that are specifically upregulated by the *acs-20* mutant under DR for lifespan extension phenotypes. Knockdown of *ptr-8*, which encodes an ortholog of the Hedgehog receptor, significantly extends lifespan of the *eat-2; acs-20* mutant. We then constructed a *ptr-8* knockout (KO) mutant via CRISPR/Cas9. In the *acs-20* mutant, DR induces significant lifespan extension in the absence of PTR-8 (*ptr-8; acs-20* vs. *ptr-8 eat-2; acs-20*) compared to the controls (*acs-20* vs. *eat-2; acs-20*), which show no significant DR-mediated changes in lifespan (Fig. 4a). In addition, RT-qPCR assays confirmed that compared to the *eat-2* mutant, the *ptr-8* mRNA levels are robustly elevated by nearly 10-fold in the *eat-2; acs-20* double mutant, whereas the difference between N2 and *acs-20* mutant is not significant (Fig. 4b). Furthermore, the *ptr-8* deletion reduces mRNA levels of ER stress-related *abu* genes in *eat-2; acs-20* compared to *eat-2* (Fig. 4c), and significantly increases the resistance to various ER stressors including tunicamycin, DTT, and thapsigargin in the *eat-2; acs-20* mutant (Fig. 4d and Supplementary Fig. 4).

Since the *ptr-8* deletion significantly abrogates the ER stress response in *eat-2; acs-20*, we then set out to characterize the role of PTR-8 in proteostasis using the transgenic polyQ model. Imaging and quantification experiments show that compared to the control RNAi, the *ptr-8* RNAi treatment significantly reduces the amount of the Q35::YFP aggregates in the *eat-2; acs-20* double mutant (Fig. 4e, f). Consistently, the *ptr-8* RNAi treatment also significantly delays age-related paralysis caused by the Q35::YFP aggregation (Fig. 4g). To make more quantitative assessment of DR, *acs-20*, and *ptr-8* for their roles in proteostasis, we performed filter trap assays and immunoblots to measure aggregated and total Q35::YFP respectively in the wild-type, *eat-2*, *acs-20*, and *eat-2; acs-20* animals treated with the control or *ptr-8* RNAi (Fig. 4h). Quantification analysis shows that compared to the wild-type animals, the DR-mimicking *eat-2* mutant has significantly less Q35::YFP aggregation, whereas this effect is abolished by the *acs-20* mutation (Fig. 4i). On the other hand, inhibition of *ptr-8* restores the protective effect of DR as indicated by significantly reduced levels of Q35::YFP aggregates in the *eat-2; acs-20* mutant treated with the *ptr-8* RNAi compared to the control RNAi (Fig. 4h, i). Altogether, these results demonstrate that PTR-8 functions downstream of *acs-20* as a key regulator of proteostasis and longevity under DR.

### NHR-23/RORA functions downstream of *acs-20* to repress *ptr-8* transcription

Since the *ptr-8* mRNA levels are significantly elevated in the *eat-2; acs-20* mutant, we constructed a reporter strain that carries a 3 kb *ptr-8* promoter driving *gfp* transgene to study the transcriptional regulation mechanisms. This reporter strain shows GFP expression mainly in the epidermis, and the epidermal GFP signal is significantly increased in the *eat-2; acs-20* mutant compared to the wild type (Fig. 5a). These expression patterns suggest that there might be either transcriptional repressors that restrict *ptr-8* expression in the wild-type background, or transcriptional activators that promote *ptr-8* expression in the *eat-2; acs-20* mutant. To test these hypotheses, we performed genetic screens for altered *ptr-8p::gfp* expression using the transcription factor RNAi sub-library in the wild-type and *eat-2; acs-20* mutant backgrounds, respectively. Although we did not find any RNAi treatment

that reduces the high level of *ptr-8p::gfp* expression in *eat-2; acs-20*, we identified NHR-23 as a transcriptional repressor of *ptr-8* in the wild type as RNAi knockdown of *nhr-23* increases not only expression levels of the *ptr-8p::gfp* reporter (Fig. 5b), but also endogenous *ptr-8* mRNA levels as quantified via RT-qPCR (Fig. 5c). *nhr-23* encodes a nuclear hormone receptor that is orthologous to RORA, a key transcriptional regulator of the circadian clock[33]. Like *acs-20*, epidermis-specific RNAi knockdown of *nhr-23* also significantly blocks DR-induced lifespan extension (Fig. 5d).

To further characterize the transcriptional regulation of *ptr-8* by NHR-23, we constructed transgenic lines carrying serial deletions of *ptr-8* promoter::*gfp* reporters. In contrary to the 3 kb promoter, the *ptr-8* p(Δ1.5-3k)::*gfp* reporter does not respond to *nhr-23* RNAi knockdown (Fig. 5e), suggesting the key *cis*-regulatory element is located within the 1.5–3 kb region of the *ptr-8* promoter. Further studies narrow the *cis*-regulatory element down to the 1.9–2.7 kb region of the *ptr-8* promoter (Fig. 5e). We then set out to investigate whether the identified promoter region is functional regarding to the regulation of *ptr-8* transcription and longevity in vivo. CRISPR/Cas9-based genome editing was performed to create a 833-bp deletion mutant of the 1.9–2.7 kb *ptr-8* promoter region. RT-qPCR experiments reveal that the significantly elevated *ptr-8* expression upon *nhr-23* or *acs-20* RNAi treatment is completely blocked in the absence of this promoter region (Fig. 5f, g). To examine whether NHR-23 interacts with the *ptr-8* promoter region, we performed genome editing experiment to create the *nhr-23::gfp* knockin strain. ChIP-qPCR using anti-GFP antibodies show that NHR-23::GFP has reduced interaction with the *ptr-8* promoter in the absence of ACS-20 (Fig. 5h). Moreover, deletion of the 1.9–2.7 kb *ptr-8* promoter region is sufficient to restore DR-induced lifespan extension when animals are subjected to *nhr-23* or *acs-20* RNAi knockdown (Fig. 5i, j).

NHR-23 does not seem to be the only ACS-20 downstream transcriptional regulator of *ptr-8* since deletion of the 1.9–2.7 kb *ptr-8* promoter region is not sufficient to increase *ptr-8* expression (Fig. 5e–g) or block DR-induced lifespan expression (Fig. 5i, j). One plausible model is that NHR-23 functions through inhibiting a transcriptional activator of *ptr-8* rather than directly represses *ptr-8* expression (Fig. 6). When the *cis*-regulatory element is deleted, neither NHR-23 or the unknown factor can affect *ptr-8* expression. Animals thus show normal response to DR. Taken together, these results demonstrate that NHR-23 functions downstream of ACS-20 to repress *ptr-8* expression and ensure DR-induced lifespan extension.

In summary, we have identified ACS-20/FATP4 as a key mediator of DR on healthy ageing from a targeted genetic screen. ACS-20 functions in the epidermis during development to regulate DR-induced lifespan extension. Under DR, ACS-20 functions through the NHR-23/RORA nuclear hormone receptor and potentially another transcriptional regulator via a *cis*-regulatory element in the *ptr-8* promoter to repress its expression, thus ensuring proteostasis and prolonged longevity (Fig. 6). These findings reveal the key role of a lipid metabolic enzyme in nutrients-mediated modulation of ageing and elucidate the underlying transcriptional regulation mechanisms.

## Discussion

DR is an effective method to delay ageing in many species. Previous studies have identified key mediators of DR-induced longevity, many of which are transcriptional regulators. However, it has not been clear whether metabolic enzymes play key roles in the response to DR. Lipid metabolism plays important roles in both health and diseases. It has been well-documented that obesity is associated with many pathologies. Intriguingly, many long-lived *C. elegans* mutants, such as *daf-2*, *rsks-1*, and *glp-1*, show increased lipid accumulation, whereas certain longevity mutant such as *eat-2*, has reduced lipid levels[34]. Lipid metabolism is a very complex and dynamic process including synthesis,

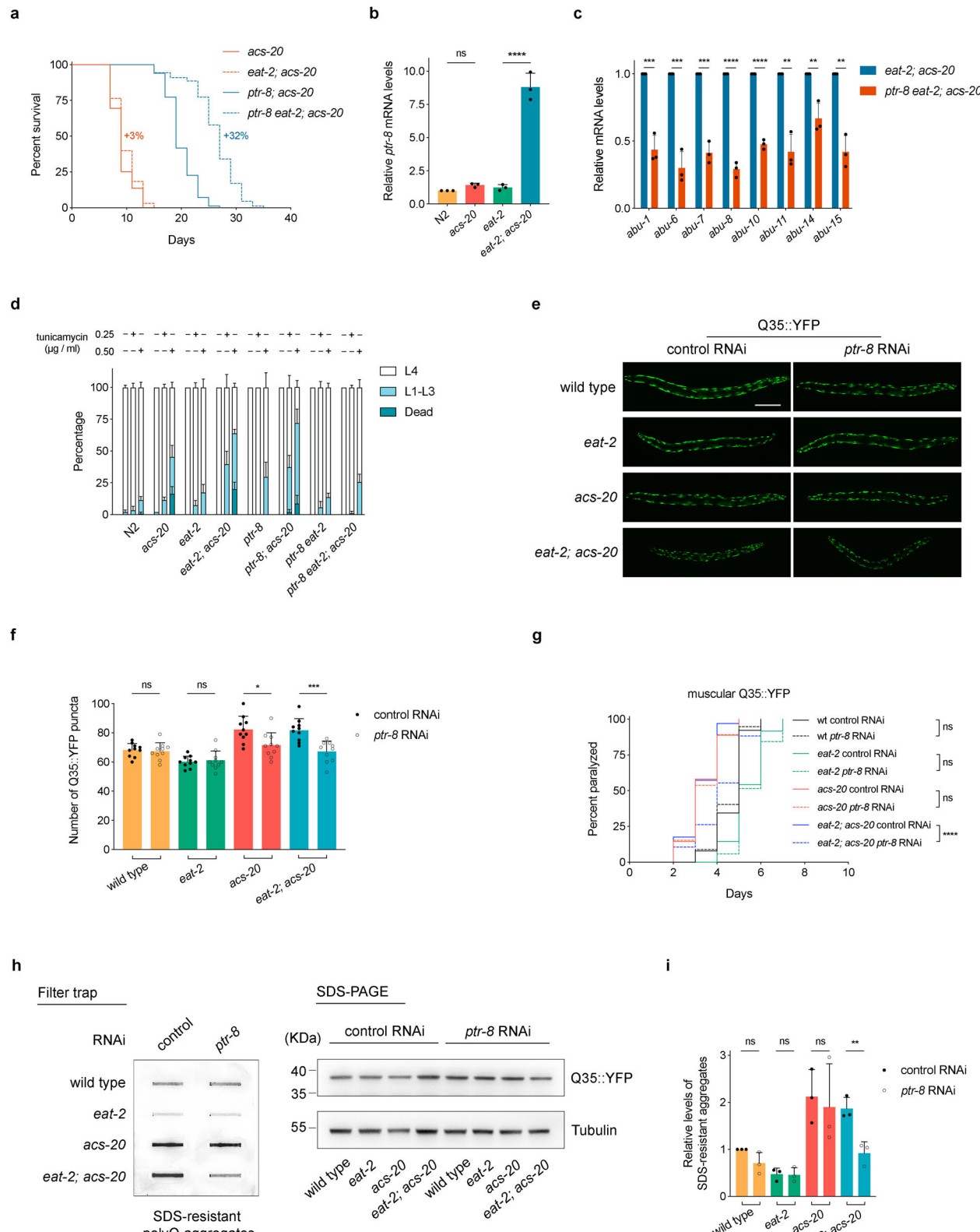

breakdown, transportation, and so on. In this study, we set out to characterize the roles of lipid metabolic enzymes in DR-induced longevity by a targeted RNAi screen of the *acs* family of genes, which are involved in the rate-limiting step of fatty acid metabolism[20]. We found that ACS-20 is required for DR-induced lifespan and healthspan extension (Fig. 1). Unlike some mediators of DR that are functional only in certain DR regimens, ACS-20 is required for the lifespan extension

by bacterial food dilution or the *eat-2* genetic mimic of DR, suggesting it is bona fide mediator of DR.

The temporospatial analysis reveals that ACS-20 functions in the epidermis during development to regulate DR-induced longevity, which is consistent with the endogenous expression patterns of ACS-20 (Fig. 2). It is intriguing that certain interventions, such as perturbation of the mitochondrial electron transport chain and removal of

**Fig. 4 | Elevated PTR-8 expression in *acs-20* disrupts the protective effect of DR on lifespan and proteostasis. a** Survival curves of *acs-20*, *eat-2; acs-20*, *ptr-8; acs-20*, and *ptr-8 eat-2; acs-20*. *acs-20* vs. *eat-2; acs-20*, *p* = 0.0995; *ptr-8; acs-20* vs. *ptr-8 eat-2; acs-20*, *p* < 0.0001 (log-rank tests). Percentages indicate changes in mean lifespan induced by *eat-2*. **b** RT-qPCR quantification of *ptr-8* mRNAs in N2, *acs-20*, *eat-2*, and *eat-2; acs-20*. Data are represented as mean ± SD (*n* = 3). ****$p$ < 0.0001; ns, *p* = 0.7886 (One-way ANOVA with Turkey's multiple comparisons test). **c** RT-qPCR quantification of *abu* genes in *eat-2; acs-20* and *ptr-8 eat-2; acs-20*. Data are represented as mean ± SD (*n* = 3). ****$p$ < 0.0001; ***$p$ < 0.001; **$p$ < 0.01 (two-tailed *t*-tests). **d** Percentages of N2, *acs-20*, *eat-2*, *eat-2; acs-20*, *ptr-8*, *ptr-8; acs-20*, *ptr-8 eat-2*, and *ptr-8 eat-2; acs-20* that died (dead), developmentally arrested (L1-L3) or completed development (L4) upon 0, 0.25 or 0.50 μg/ml tunicamycin treatment. Data are represented as mean ± SD (*n* = 3). For animals that completed development upon tunicamycin treatment, *eat-2* vs. *eat-2; acs-20*, *p* < 0.0001; *eat-2; acs-20* vs. *ptr-8 eat-2; acs-20*, *p* < 0.0001 (Two-way ANOVA with Turkey's multiple comparisons test). **e, f** Representative photographs of wild-type, *acs-20*, *eat-2*, and *eat-2; acs-20* animals expressing muscular Q35::YFP treated with the control or *ptr-8* RNAi (**e**) and quantification of the Q35::YFP punctae (**f**). Data are represented as mean ± SD (*n* = 10). ****$p$ < 0.0001; **$p$ < 0.01 (One-way ANOVA with Turkey's multiple comparisons test). Scale bar, 100 μm. **g** Paralysis analysis of the wild-type, *eat-2*, *acs-20*, and *eat-2; acs-20* mutant animals expressing muscular Q35::YFP with control or *ptr-8* RNAi. ****$p$ < 0.0001; ns, *p* > 0.05 (log-rank tests). **h, i** Immuno-blots (**h**) and quantification (**i**) of the SDS-resistant Q35::YFP aggregates, total Q35::YFP, and tubulin in the wild-type, *eat-2*, *acs-20*, and *eat-2; acs-20* with control or *ptr-8* RNAi. Data are represented as mean ± SD (*n* = 3). **$p$ < 0.01; ns, *p* > 0.05 (One-way ANOVA with Turkey's multiple comparison tests). Source data are provided as a Source Data file.

the germline via physical or genetic methods, are required to be performed during development to affect *C. elegans* ageing[35,36]. The *acs-20* deletion mutant does not cause developmental arrest or sterility, suggesting it might not cause sickness in general. The epidermal tissue not only serves as a barrier for protection, but also plays important roles in body morphology and physiology, such as secretion, lipid storage, glia-like functions, and innate immunity[37,38]. It has been reported that reduced insulin-like signaling functions through the SKN-1 transcription factor to delay ageing via collagen genes, the expression products of which are secreted by epidermal cells to form the cuticle in *C. elegans*[39]. Here we found *acs-20* and the downstream genes all function in the epidermis to regulate the effect of nutrients on longevity, which highlights the importance of this tissue for further exploration in ageing research.

Transcriptome profiling reveals that genes upregulated in the *eat-2; acs-20* double mutant are enriched with UPR$^{ER}$-related genes, and the *acs-20* mutant is more sensitive to the ER stress-inducing drugs tunicamycin, DTT and thapsigargin (Fig. 3 and Supplementary Fig. 4). These results suggest that the *acs-20* mutant might have disrupted proteostasis, which has been associated with ageing and age-related degenerative diseases[40]. To further explore the underlying molecular mechanisms, we performed an RNAi-based genetic screen of genes upregulated in the short-lived *eat-2; acs-20* double mutant for the lifespan extension phenotype. PTR-8, a *C. elegans* ortholog of Patched from the Hedgehog pathway, was identified as a key downstream regulator of lifespan and proteostasis (Fig. 4). The Hedgehog pathway plays important roles during development. However, this signaling has certain key components missing but significant expansion of Hedgehog receptor genes in *C. elegans*[41]. Whether this non-canonical function of PTR-8/Patched in healthy ageing is conserved in higher organisms should be explored in future studies.

Since *ptr-8* is transcriptionally elevated in the *eat-2; acs-20* double mutant, we performed a targeted RNAi screen of 757 transcription factor-related genes and identified NHR-23 as a transcriptional repressor of *ptr-8*. Consistent with the gene expression results, RNAi knockdown of NHR-23 in the epidermis also blocks DR-induced longevity (Fig. 5). NHR-23 is orthologous to the nuclear receptor RORA, which plays an important role in circadian clock regulation. It has been reported that inhibition of key regulators of the circadian clock blocks DR-induced longevity through affecting lipid turnover in *Drosophila*[42]. Although *C. elegans* does not have a 24-h rhythm, certain circadian clock genes are conserved, and they are involved in the regulation of developmental timing in the epidermis, such as molting. Our findings highlight the importance of circadian clock genes in DR-induced healthy ageing.

To further dissect the transcriptional regulation of *ptr-8* by NHR-23, we mapped the *cis*-regulatory element to a specific region of the *ptr-8* promoter. Although NHR-23 directly binds to this region, gene expression studies suggest that NHR-23 might function through another transcriptional regulator to affect *ptr-8* expression indirectly (Fig. 5 and Fig. 6). It will be interesting to identify this unknown transcriptional regulator and further elucidate the mechanisms of gene expression in future studies.

Does ACS-20 play regulatory roles in other long-lived mutants? To address this question, we crossed the *acs-20* mutation into the long-lived *daf-2* (IGF-1 receptor), *glp-1* (Notch receptor), *rsks-1* (ribosomal S6 kinase), and *isp-1* (iron-sulfur protein of the mitochondrial ETC complex III) mutants, which represent reduced insulin/IGF-1 signaling[43,44], germline deficiency[36], reduced TOR signaling[45], and perturbation of the mitochondrial electron transport chain[46], respectively. Survival assays show that the *acs-20* mutation partially blocks the lifespan extension by *daf-2* and *glp-1*, whereas it completely suppresses the prolonged longevity by *rsks-1* or *isp-1* (Supplementary Fig. 5). These data suggest that the effects of *acs-20* on longevity are context-dependent, and the suppression of longevity by *acs-20* is unlikely due to sickness in general.

The relationship between lipid metabolism and longevity is complicated. Both increased and decreased lipid accumulation were originally associated with prolonged longevity, but more and more evidence indicate that it is not the level of lipids but the active utilization of lipids that contributes to the lifespan extension[34]. Our lipid staining and lipid droplets GFP reporter assays show that the *acs-20* mutation does not have a strong effect on neutral lipid levels in the intestine or epidermis (Supplementary Fig. 3). Further biochemical analysis shows that inhibition of *acs-20* does not change the levels of most fatty acids, but it does affect certain phospholipids and sphingolipids (Supplementary Fig. 3e-g). It was reported that ACS-3, another long-chain acyl-CoA synthetase, functions in epidermal cells to direct the biosynthesis of phospholipid ligands for the NHR-25 nuclear receptor in *C. elegans*[47]. It will be interesting to determine whether certain lipid metabolic products of ACS-20 function as the ligand for NHR-23 to regulate *ptr-8* expression, proteostasis, and longevity in future studies.

Our findings demonstrate that ACS-20, NHR-23, and PTR-8 all function in the epidermis to regulate proteostasis and longevity under DR conditions. Since the polyQ model used to assess proteostasis is expressed in muscle cells and the lifespan phenotypes occur at the whole animal level, it is plausible that these regulators might work through cell-non-autonomous manner to affect ageing. It will be interesting to further analyze the molecular events downstream of PTR-8 via genetic and genomic approaches to better understand how non-canonical Hedgehog signaling affects ageing in response to nutrients.

In summary, we have identified a fatty acid metabolic enzyme as the key modulator of healthy ageing in response to nutrients. The underlying mechanisms involve transcriptional regulation of the Hedgehog signaling and maintenance of proteostasis. Further characterization of the molecular details should help to better understand the mechanisms of ageing and develop pharmaceutical interventions to delay ageing.

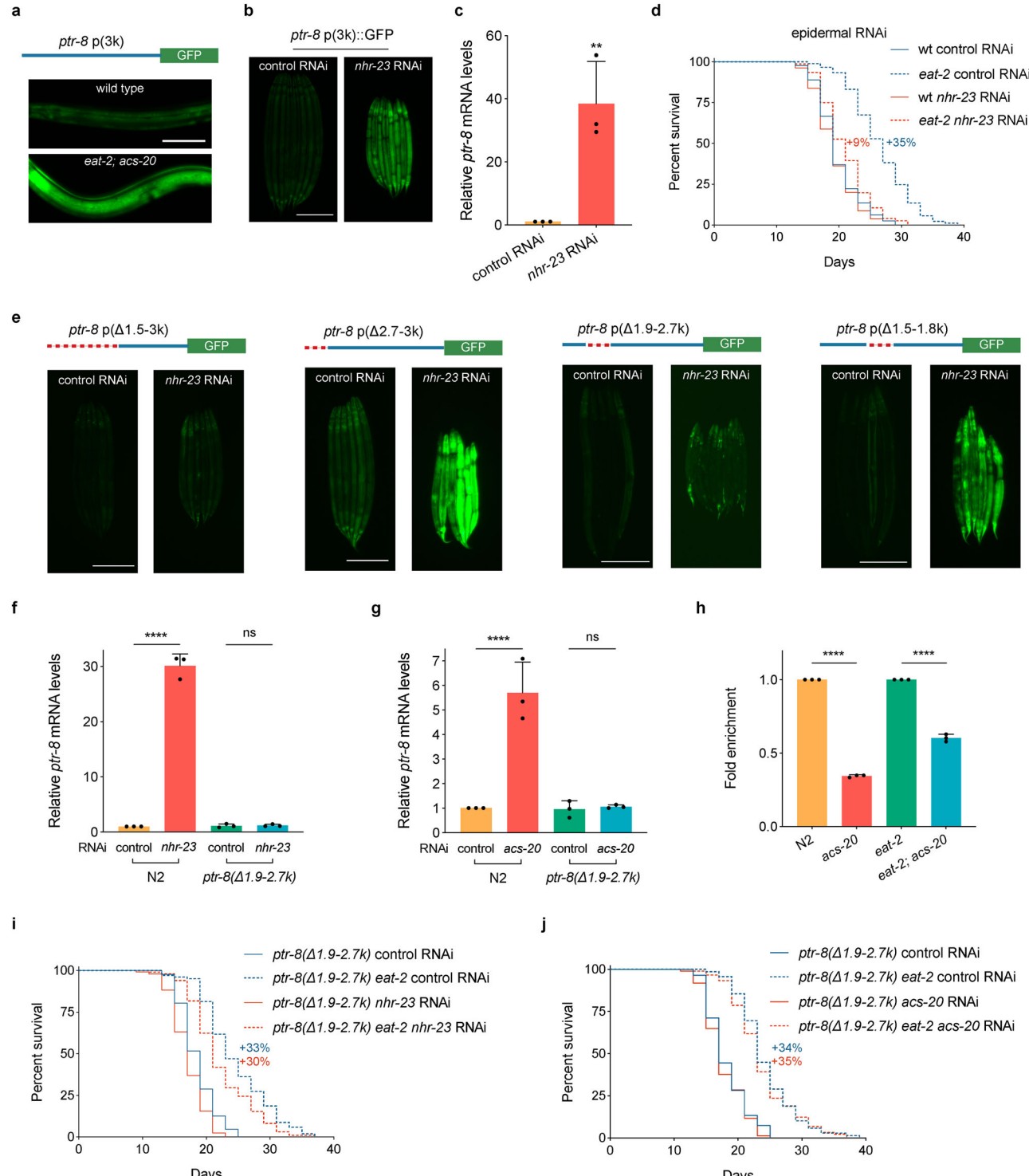

**Fig. 5 | NHR-23/RORA mediates the effect of *acs-20* on *ptr-8* expression and lifespan. a** Representative photographs of the *ptr-8* p(3k)::GFP reporter expression in wild-type and *eat-2; acs-20* animals. Scale bar, 50 μm. **b** Representative photographs of the *ptr-8* p(3k)::GFP expression in animals treated with the control or *nhr-23* RNAi. Scale bar, 200 μm. **c** RT-qPCR quantification of *ptr-8* mRNA levels in wild-type animals treated with the control or *nhr-23* RNAi. Data are represented as mean ± SD (*n* = 3). **\*\*p* = 0.0084 (two-tailed *t*-test). **d** Survival curves of wild type (wt) and *eat-2* mutant animals treated with the epidermis-specific control or *nhr-23* RNAi. wt vs. *eat-2*: control RNAi, *p* < 0.0001; *nhr-23* RNAi, *p* < 0.01 (log-rank tests). **e** Representative photographs of the *ptr-8* p(Δ1.5-3 k)::GFP, *ptr-8* p(Δ2.7-3 k)::GFP, *ptr-8* p(Δ1.9-2.7 k)::GFP, and *ptr-8* p(Δ1.5-1.8 k)::GFP expression in animals treated with the control or *nhr-23* RNAi. Scale bar, 200 μm. **f, g** RT-qPCR quantification of *ptr-8* mRNA levels in N2 and the *ptr-8* p(Δ1.9-2.7 k) mutant animals treated with

either control vs. *nhr-23* RNAi (**g**) or control vs. *acs-20* RNAi (**h**). Data are represented as mean ± SD (*n* = 3). **\*\*\*\*p* < 0.0001; ns, *p* = 0.9972 (*nhr-23* RNAi), *p* = 0.9972 (*acs-20* RNAi) (One-way ANOVA with Turkey's multiple comparisons test). **h** ChIP-qPCR quantification of the relative enrichment of NHR-23::GFP binding with the *ptr-8* promoter region (1861–1958 bp) in N2, *acs-20*, *eat-2* and *eat-2; acs-20*. The fold enrichment was normalized by the input DNA. Data are represented as mean ± SD (*n* = 3). \*\*\*\*, *p* < 0.0001 (two-tailed *t*-tests). **i, j** Survival curves of *ptr-8* p(Δ1.9-2.7 k) and *ptr-8* p(Δ1.9-2.7 k) *eat-2* mutant animals treated with the control vs. *nhr-23* RNAi (**i**) or *acs-20* RNAi (**j**). *ptr-8* p(Δ1.9−2.7 k) vs. *ptr-8* p(Δ1.9−2.7 k) *eat-2*: control RNAi, *p* < 0.0001; *nhr-23* RNAi, *p* < 0.0001; *acs-20* RNAi, *p* < 0.0001 (log-rank tests). Percentages indicate changes in mean lifespan induced by the *eat-2* mutant. Source data are provided as a Source Data file.

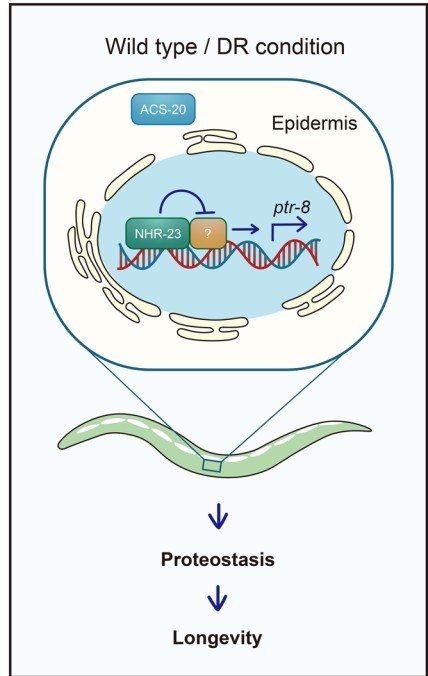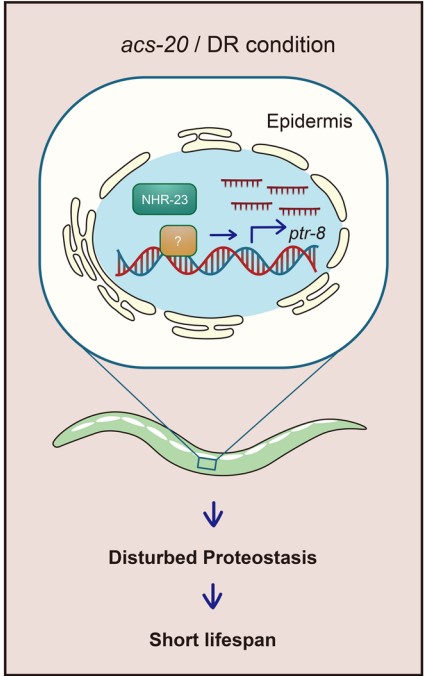

**Fig. 6 | Schematic representation on the regulation of DR-induced healthy ageing by ACS-20, NHR-23, and PTR-8.** ACS-20 functions in the epidermis during development to regulate DR-induced lifespan extension. In the presence of ACS-20, the NHR-23/RORA nuclear hormone receptor functions with an unknown factor to restrict the expression of PTR-8/Patched via a *cis*-regulatory element to maintain proteostasis and promote healthy ageing under DR. In the absence of ACS-20, NHR-23-mediated transcriptional repression of *ptr-8* is abolished. Overexpression of PTR-8 results in disrupted proteostasis and elevated ER stress, which block the lifespan extension produced by DR.

## Methods

### C. elegans strains
All strains were maintained on NGM agar plates seeded with *E. coli* OP50 at 20 °C. A list of *C. elegans* strains used in this study can be found in Supplementary Data S2.

### Lifespan assay
Lifespan assays were performed at 20 °C unless otherwise stated. To prevent progeny production, 20 μg/ml (+)-5-fluorodeoxyuridine (FUdR) was added onto NGM plates during the reproductive period (Day 1–7 of adulthood). The first day of adulthood is day 1 on survival curves. Animals were scored as alive, dead, or lost every other day. Animals that did not respond to gentle touch were scored as dead. Animals that died from causes other than ageing, such as sticking to the plate walls, internal hatching or bursting in the vulval region, were scored as lost. Lifespan assays under differential nutrient conditions were performed using the modified solid DR method[14]. Day 1 adult animals were treated with bacterial food at different concentrations and survival was monitored at 25 °C. Kaplan–Meier survival curves were plotted for each lifespan assay, and statistical analyses (log-rank tests).

### RNAi by feeding
RNAi experiments were performed by feeding worms *E. coli* strain HT115 (DE3) transformed with either the empty vector L4440 as the control or gene-targeting constructs from the *C. elegans* Ahringer RNAi Collection[48]. Overnight bacterial culture in LB supplemented with Ampicillin (100 μg/ml) at 37 °C was seeded onto NGM plates containing IPTG (1 mM) and Ampicillin (100 μg/ml) and incubated at the room temperature overnight to induce double-stranded RNAs production. Embryos or L1 larvae were placed on RNAi plates and incubated at 20 °C until L4 or adulthood to score phenotypes. For *nhr-23*, the RNAi treatment was initiated at the L2 stage to avoid sickness produced by *nhr-23* knockdown.

### Stress resistance assays
For the heat stress assays, synchronized day 1 adult worms were incubated at 35 °C for 10 h before counting the numbers of alive or dead animals. For the ER stressor tolerance assays, gravid adults were put on the plates supplemented with different concentrations of tunicamycin (0, 0.25, 0.50 μg/ml), DTT (0, 25, 50 μg/ml) or thapsigargin (0, 25, 50 μg/ml) to lay eggs for 1 h. The developmental stages of the progeny were quantified when most animals from the control group have reached to the L4 stage or beyond.

### Fluorescence imaging of *C. elegans*
Animals were anesthetized in 1% sodium azide and immediately imaged. For the *ptr-8* transcriptional reporters, animals were imaged with a Leica MC165 FC dissecting microscope equipped with a Leica DFC450 C digital camera. For ACS-20::GFP or Q35::YFP, animals were imaged with a Zeiss LSM880 confocal microscope equipped with a 40X/1.3 W objective controlled by the ZEN software (Carl Zeiss). For GFP or YFP, a 488 nm laser was used for excitation, and signals were collected with a 500-550 nm emission filter. In all the imaging studies, images within the same figure panel were taken with the same exposure time and adjusted with identical parameters using Adobe Photoshop or ImageJ.

### Western blot and antibodies
Roughly equal numbers of synchronized L4 animals were manually transferred into the lysis buffer (150 mM NaCl, 1 mM EDTA, 0.25% SDS, 1.0% NP-40, 50 mM Tris-HCl [pH7.4], Roche complete protease inhibitors and phosSTOP phosphatase inhibitors) supplemented with the 4 × SDS loading buffer and immediately frozen at -80 °C. Samples were boiled for 10 min before resolving on precast SDS-PAGE gels (GenScript). Bands of interests were quantified using the ImageJ software and normalized to the intensities of α-tubulin, the internal control. Antibodies used in Western blots include monoclonal anti-FLAG

(Sigma, 1804) and monoclonal anti-Tubulin Alpha (Sigma, T6074), which are both used at 1:10,000 dilution.

## Filter trap assays

Synchronized Day 3 adult worms were collected with the M9 buffer, and immediately frozen with liquid nitrogen. Worm pellets were thawed on ice, and worm extracts were generated by sonication in the FTA buffer [50 mM Hepes (pH 7.4), 1 mM EDTA, 150 mM NaCl, and 1% Triton X-100] supplemented with the EDTA-free protease inhibitor cocktail (Roche). Worm debris was removed by centrifugation at 8000 g for 15 min at 4 °C. 100 µg of protein extracts were supplemented with SDS at the final concentration of 0.5% and loaded onto a cellulose acetate membrane with 0.22 µm pore size assembled in a slot blot apparatus (Bio-Rad). The membrane was then washed with 0.2% SDS and SDS-resistant protein aggregates were assessed by immunoblotting using antibodies against GFP (AMSBIO, TP401, 1:500 dilution). Extracts were also analyzed by SDS-PAGE and Western blots with the anti-GFP (AMSBIO, TP401, 1:500 dilution) and anti-Tubulin Alpha (Sigma, T6074, 1:10,000 dilution) primary antibodies, as well as the goat anti-mouse IgG (H + L) HRP secondary antibodies (Bioworld BS12478, dilution 1:10,000). The intensities of bands from filter trap and Western blot were measured with ImageJ.

## mRNA-Seq and bioinformatics analysis

Three biological replicates of total mRNAs from N2, *eat-2*, *acs-20*, and *eat-2; acs-20* mutant L4 larvae were sent to Berry Genomics for library construction and pair-ended sequencing with the length of 100 nucleotides, 11.5 million reads per sample on a HiSeq2000 machine (Illumina). Reads were aligned to the *C. elegans* genome (WS220) using the spliced-junction mapper TopHat2[49]. Aligned reads were counted per gene using the python script HTseq[50]. Differentially expressed genes were determined via DESeq2. The mRNA-Seq data have been deposited at the NCBI under accession number GSE125718.

## GO analysis

GO analyses was performed using Metascape (https://metascape.org/). Terms with a $p < 0.01$, a minimum count of 3, and an enrichment factor >1.5 were collected and grouped into clusters based on their membership similarities[51].

## RT-qPCR

Synchronized L4 animals were collected for total RNA extractions using the Trizol reagent (Takara) and Direct-zol RNA mini prep kit (ZYMO Research). The cDNA was synthesized by the reverse transcription system (Takara). The SYBR Green dye (Takara) was used for qPCR reactions carried out in triplicates on a Roche LightCycler 480. Relative gene expression levels were calculated using the $2^{-\Delta\Delta Ct}$ method[52]. RT-qPCR experiments were performed at least three times with consistent results using independent RNA preparations.

## CRISPR/Cas9 alleles generation

CRISPR engineering with a self-excising drug selection cassette (SEC) were performed to knock-in GFP::3 × FLAG at the C-terminal of ACS-20 and NHR-23 using the homologous recombination approach[53]. The injection mix contained two plasmids that drive expression of two different Cas9-sgRNAs (50 ng/ml), a selection marker pCFJ90 (P*myo-2::mCherry::unc-54-3'*-UTR) (2.5 ng/ml, Addgene #19327) and a homologous recombination plasmid (50 ng/ml). To generate the sgRNA plasmids, primers were designed with the CRISPR DESIGN tool (https://zlab.bio/guide-design-resources) and inserted into the pDD162 vector (Addgene #47549) using the site-directed mutagenesis kit (TOYOBO SMK-101). The ccdB sequences from the pDD282 vector (Addgene #66823) were replaced with two homologous arms (500–700 bp) to generate the repair template FP-SEC plasmid. Injected animals and their progeny were treated with Hygromycin B

(350 mg/ml) to select successful knock-in events. Hygromycin B resistant roller animals were tested by PCR genotyping. The SEC was removed by heat shock and homozygous knock-in alleles were confirmed by PCR and DNA sequencing.

## ChIP-qPCR

Synchronized L4 larvae were washed with the M9 buffer for three times and fixed with 37% formaldehyde for 30 min followed by stopping with 2.5 M Glycine for 15 min. Samples were washed twice with 1 X PBS, once with the CL-WS1 buffer [50 mM HEPES/KOH pH 7.5, 150 mM NaCl, 0.1% sodium deoxycholate, 1% Triton X-100, 1 mM EDTA, 1 mM PMSF, EDTA-free protease inhibitor cocktail], ground in liquid nitrogen, resuspended in 2 ml ice-cold CL-Lysis buffer [50 mM HEPES-KOH pH 7.5, 150 mM NaCl, 0.1% sodium deoxycholate, 1% Triton X-100, 0.1% SDS, 1 mM EDTA, 1 mM DTT, 1 mM PMSF, EDTA-free protease inhibitor cocktail], and sonicated using a Bioruptor UCD-200 at high mode for 15 cycles of 30 s on and 30 s off. Samples were then centrifuged at 18,000 g for 15 min at 4 °C, and the supernatant was transferred to a new tube. 5% of the extracts were saved as input sample, and 5 µg of chromatin as ChIP samples were incubated with GFP-Trap Dyna beads (ChromoTek, # gtd) for 1 h, and then washed twice with the CL-WS2 buffer [50 mM HEPES/KOH pH 7.5, 1 M NaCl, 0.1% sodium deoxycholate, 1% Triton X-100, 1 mM EDTA, 1 mM PMSF, EDTA-free protease inhibitor cocktail], once with the CL-WS3 buffer [10 mM Tris·HCl pH 8.0, 0.25 M LiCl, 1% sodium deoxycholate, 1% NP-40, 1 mM EDTA, 1 mM PMSF, EDTA-free protease inhibitor cocktail], once with the CL-WS4 buffer [10 mM Tris·HCl, pH 8.0, 1 mM EDTA]. 100 µl of the CL-Elution buffer [10 mM Tris·HCl, pH 8.0, 250 mM NaCl, 1% SDS, 1 mM EDTA] was added to the sample and incubated for 30 min at 65 °C. ChIP samples were vortexed, and the supernatants were collected. RNaseA was then added to both the ChIP and input samples, and incubated for 1 h at 37 °C. The crosslink was reversed with proteinase K at 65 °C for >4 h. DNA was purified with the Qiagen Maxtract kit (Qiagen, #129046) and used as templates for qPCR.

## Oligo sequences

A list of oligo sequences used for qPCR and CRISPR/Cas9 experiments in this study can be found in Supplementary Data S3.

## Statistical analysis

All statistical analyses were preformed using two-tailed *t*-tests, one-way or two-way ANOVA with Turkey's multiple comparisons test, and log-rank tests as reported in figure legends and Source Data. Bar graphs are presented as mean ± SD with $p < 0.05$ considered as significant differences.

## Reporting summary

Further information on research design is available in the Nature Portfolio Reporting Summary linked to this article.

# Data availability

All data are included either in the paper and or in the supplementary information. The mRNA-Seq data have been deposited at the NCBI under accession number GSE125718. There are no restrictions on data availability. Source data are provided with this paper.

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

## Acknowledgements

We thank Drs. Qian Bian, Shiqing Cai, Mengqiu Dong, Bin Liang, Ho Yi Mak, Shohei Mitani, Guangshuo Ou, Jarod A. Rollins, Billy Qi, Xiong Su, Shaobing Zhang, and Zhiwen Zhu for resources, advice, and discussion. We thank Mrs. Yanling Bo, Fen Chen, Xuan Zhang and Mr. Yijie Jin for technical assistance. Some strains were provided by the CGC and the Japanese National BioResource Project. This work was supported by grants from the National Key R&D Program of China (2021YFA0805800) and National Natural Science Foundation of China (32171156, 31971092, 31671527) to D.C.

## Author contributions

D.C. conceived the study. Z.Wa., L.Z., Y.Z., M.Z., S.Z., D.W., J.L., X.Z., Q.W., Ha.Z., and Z.Wu carried out the experiments. Z.Wa., Hu.Z., and D.C. designed the experiments and interpreted the results. Z.Wa. and D.C. wrote the paper.

## Competing interests

The authors declare no competing interests.
