## [Peer Review File · Nature Communications]

ACS-20/FATP4 mediates the anti-ageing effect of dietary restrictionREVIEWER COMMENTS

Reviewer #1 (Remarks to the Author):

REVIEWER COMMENTS

In this manuscript, the authors dissect a dietary restriction (DR)-dependent anti-aging pathway that acts through the fatty acid transporter ACS-20. First, they identified that the lack of *acs-20* reduced the lifespan of animals subjected to DR more than any other *acs* genes. Next, they show that ACS-20 is required for healthspan extension. They also show that ACS-20 is mostly expressed in the epidermis during development and that it is required during development to extend DR-dependent lifespan. Next, they identified most of the *abu* genes were upregulated in the absence of *acs-20* during calorie restriction. These genes are known to be upregulated upon the endoplasmic reticulum (ER) stress response called the unfolded protein response (UPR). From the bulk RNA-seq data, they claimed that *acs-20* mutation causes ER stress under DR. They also concluded that ACS-20 protects cytosolic proteostasis which is dependent on the Hedgehog receptor ortholog PTR-8. Finally, they concluded that the regulation of ACS-20/PTR-8 axis is modulated by the transcription regulator NHR-23.

Overall, the manuscript is comprehensive with a logical flow. In general, the quality of the reported experimental data is good and well presented. However, some of the authors' conclusions are not supported by the reported data. I recommend the authors to perform additional experiments to support the authors conclusions related to ER stress and cytosolic proteostasis.

Major points to address

(1) The authors concluded that the lack of *acs-20* induces ER stress under DR. However, linking *abu* genes that are known to be upregulated during the UPR and that *acs-20* mutant is more sensitive to ER stress-inducing drug tunicamycin (Tm) is insufficient to make such claims. As ER stress induces the UPR pathway, the authors should characterize the activation of at least one branch of the UPR and/or conduct assays to demonstrate that the ER is under

stress. There should be an accumulation of unfolded protein and/or aggregates. It is also possible that ER stress is induced by lipid perturbation at the ER as ACS-20 regulates lipids.

(2) Related to the major point #1, showing that tunicamycin delays development in *acs-20* animals is not sufficient to link *acs-20* to ER stress. As tunicamycin blocks N-glycosylation in the ER, many ER-resident proteins, proteins trafficked to other organelles, and secreted proteins will fail to fold in the ER, and they might be non-functional if they leave the ER. The ER of animals exposed to Tm will not be the only cell compartment affected but also other organelles part of the secretory pathway, the plasma membrane, and even the extracellular matrix will be negatively affected. Therefore, it cannot be excluded that the absence of *acs-20* together with Tm exacerbates the development of the animal independently of the ER state.

(3) The authors also link cytosolic proteostasis to PRT-8 and ACS-8. The data is insufficient to make such a conclusion. The authors based their conclusion on (1) PRT-8 exacerbates Tm-induced development defects and (2) cytosolic polyQ accumulation is reduced in the absence of *acs-20* and *ptr-8* during DR. As described in major point #2, it is difficult to directly link Tm with proteostasis so the authors should be careful in their interpretation or to conduct more experiments. The difference in polyQ accumulation is interesting but there is a missing link of how the absence of both genes prevents the accumulation of aggregates.

(4) The authors proposed a model where ACS-20 inhibits the expression of *ptr-8* via the transcription modulator NHR-23 and that in turn promotes proteostasis and longevity. Based on their results, the links between each player are vague and I was anticipating getting more mechanistic details from this study. Although their data convincingly demonstrates that the pathway is only relevant during DR, it is not clear to me how these two are linked and what is the mechanism or cellular differences that will make their proposed pathway essential only during DR. Finally, it is also not clear why and how the role of ACS-20 during development is important to modulate lifespan. These different points should be addressed to increase the impact of the manuscript.

Minor points to address

(1) I suggest the authors to report replicates in the reported bar chart. The number of samples and replicates should be indicated in figure legends.

Reviewer #2 (Remarks to the Author):

This manuscript describes a new mechanism by which dietary restriction (DR) extends life and health span in *C. elegans*. Specifically, loss of the lipid metabolism gene *acs-20* prevents life span extension by various different ways of achieving DR, while expression of *acs-20* in the hypodermic, one of its sites of natural expression, restores life span of the DR model *eat-2*. RNA-seq analysis revealed that proteostasis genes were induced in *eat-2; acs-20* double mutants, and these mutants showed increased sensitivity to the tunicamycin, suggesting disturbed proteostasis. To identify mechanisms, the authors then tested deregulated genes in an RNAi screen and identified *ptr-8*, whose RNAi or mutation restored the extended life span of *eat-2; acs-20*, pinpointing increased *ptr-8* expression as an inhibitor of life and health span. Finally, using another RNAi screen, the authors identify and validate by promoter GFP, qPCR, ChIP and promoter bashing the Nuclear Receptor NHR-23 as a negative regulator of *ptr-8* expression, and show the deletion of a promoter element alone is sufficient to restore the long life span of *eat-2*; they also map *ptr-8* action downstream of *acs-20*.

This is a very impressive manuscript and discovery overall. The mechanism is to my knowledge new and should be of great interest to scientists studying gene regulation, life span, and metabolism. The data are very well done - conclusions are mostly very well ascertained with several different DR regimens, RNAi and mutants, tissue specific RNAi and rescue, RNA-seq and qPCR, and both life span and health span, at least with a Q35 aggregation reporter and analysis, are analyzed. The experiments are mostly well controlled and analyzed, with a couple minor issues on controls and statistical comparisons.

My only real concern revolves around the regulation of *ptr-8*. There seems to be some inconsistency with the description/rationale, and with the proposed mechanism.

1. Rationale: I'm confused how *ptr-8* was found. The authors write on line 224 "The 95

genes that are significantly up-regulated in the eat-2; acs-20 vs. eat-2 comparison but not in the acs-20 vs. N2 comparison....” and later study these 95 genes by RNAi, finding ptr-8. However, Fig 4b clearly shows the ptr-8 doesn't belong in this category since it is clearly quite strongly and significantly induced in acs-20 vs. N2. Did the RNA-seq not show this regulatory pattern? This would be a concern, especially considering the very strong and significant upregulation detected by qPCR in Fig 4d. Please show RNA-seq CPM or FPKM plots for ptr-8 for comparison with all 4 samples, with replicate data. The authors also later on line 301 state "significantly elevated ptr-8 expression upon nhr-23 or acs-20 RNAi...", so clearly acs-20 loss on its own causes ptr-8 upregulation. This is concerning as the authors' own data and argumentation appear incongruent. Or was there another reason why ptr-8 was studied?

2. Mechanism: I struggled to comprehend the proposed mechanism for NHR-23 action. Specifically, I don't understand how NHR-23 and the element in the ptr-8 promoter act on ptr-8 expression. Figure 5 shows that NHR-23 is required to repress ptr-8, as its RNAi induces the ptr-8 promoter::GFP reporter and endogenous ptr-8 mRNA, and hence depletion of NHR-23 shortens lifespan of eat-2 (wherein ACS-20 is active to repress ptr-8, dependent on nhr-23). So far so good. But then why does deletion of the element, through which NHR-23 is supposed to act, on its own not cause increased ptr-8 expression? If the element were deleted, then NHR-23 would not be able to bind, and that should cause strong derepression, resembling nhr-23 loss. Why does this not happen? Along these lines, if NHR-23 represses a gene through an element in said gene's promoter, removal of both of these factors would not be expected to have opposite effects as is the case in Fig 5f. Perhaps their model is incomplete? An alternative model could be that NHR-23 prevents another TF, an activator, from acting through this element; this would also make sense of the ChIP data - if NHR-23 binding is reduced, then the other factor X might be more active, promoting increased ptr-8 expression and shorter life span. Such transcriptional repression of other TF's activities has been well described for the glucocorticoid receptor and NFkB, with GR exerting its anti-inflammatory effects by repressing NFkB's activation function. But a direct, repression-linked function for NHR-23 doesn't appear possible to me. I'd be very interested to hear the authors response to this comment.

There are also some issues with missing controls where not all conditions have been

assessed:

3. Fig 2h is missing controls; it would be ideal to include *acs-20* and *eat-2*; *acs-20* mutants. This would be important b/c it would inform on whether transgenic *acs-20* expression can rescue *acs-20* mutation; in fact, the authors should also show the experiment where they rescued *acs-20* with a construct using its own promoter besides *dpy-7*; showing that *acs-20* overexpression on its own promotes longevity would be relevant.
4. Ideally, in Fig 4d, additional controls (*ptr-8* single mutant and doubles with *acs-20* and *eat-2*) would be added but I appreciate that this may not be possible after the fact.
5. Similar comment for Fig 4e-g, it would be ideal to have additional controls (WT, *eat-2*, *acs-20* singles with both control and *ptr-8* RNAi), but this may not be possible in revision.

Other

6. Fig 1c, e, 4b, etc for similar figures, please use appropriate ANOVA tests with multiple comparison corrections to assess all relevant comparisons, not just the indicated pairs.
7. For RNA-seq data visualization, perhaps consider a Venn diagram, which might better highlight the 95 genes of interest. Also, why were downregulated genes not considered further?
8. Methods, please add details and references on how GO term analysis was done.

Reviewer response summary

We thank both reviewers' insightful comments on our manuscript. In this study, we identified the highly conserved ACS-20/FATP4 as a key mediator of dietary restriction (DR)-induced healthy ageing in *C. elegans*. Using functional genomics, genome editing, and unbiased genetic screening approaches, we demonstrate that in the *acs-20* mutant, transcriptional dysregulation of PTR-8/Patched leads to defects in proteostasis and ER stress that restrict DR-induced lifespan and healthspan extension.

During the revision, we have acquired new experimental evidence to strengthen the manuscript. The important new discoveries and changes that we have made are summarized as follows:

- 1) We have provided new experimental evidence to demonstrate that knockout of *acs-20* abolishes activation of ER^{UPR} (Fig. 3e, f) thus making animals more sensitive to various ER stressors (Fig. 4d and Supplementary Fig. 4).
- 2) We have added more control groups based on both Reviewers' suggestions (Fig. 2h, Fig. 4d-i, Supplementary Fig. 2 and Fig. 4).
- 3) We have revised the working model of NHR-23-mediated transcriptional regulation of *ptr-8* based on Reviewer 2's suggestion (Fig. 6, Results and Discussion).
- 4) We have improved data presentation by showing replicates in graphs, reporting sample sizes in figure legends, performing more rigorous statistical analyses, and including all the quantitative and statistical results in the revised Supplementary Table 1.

Below we outlined our responses to individual reviewer's comments, which were italicized. We highlighted revisions in the main text and supplemental materials with blue font.

Individual Reviewer Responses:

Reviewer #1 (Remarks to the Author):

REVIEWER COMMENTS

*In this manuscript, the authors dissect a dietary restriction (DR)-dependent anti-aging pathway that acts through the fatty acid transporter ACS-20. First, they identified that the lack of *acs-20* reduced the lifespan of animals*

subjected to DR more than any other acs genes. Next, they show that ACS-20 is required for healthspan extension. They also show that ACS-20 is mostly expressed in the epidermis during development and that it is required during development to extend DR-dependent lifespan. Next, they identified most of the abu genes were upregulated in the absence of acs-20 during calorie restriction. These genes are known to be upregulated upon the endoplasmic reticulum (ER) stress response called the unfolded protein response (UPR). From the bulk RNA-seq data, they claimed that acs-20 mutation causes ER stress under DR. They also concluded that ACS-20 protects cytosolic proteostasis which is dependent on the Hedgehog receptor ortholog PTR-8. Finally, they concluded that the regulation of ACS-20/PTR-8 axis is modulated by the transcription regulator NHR-23.

Overall, the manuscript is comprehensive with a logical flow. In general, the quality of the reported experimental data is good and well presented. However, some of the authors' conclusion are not supported by the reported data. I recommend the authors to perform additional experiments to support the authors conclusions related to ER stress and cytosolic proteostasis.

We thank Reviewer 1 for the positive assessment of our study. We have carried out multiple experiments and revised the manuscript to address the Reviewer 1's concerns.

Major points to address

(1) The authors concluded that the lack of acs-20 induces ER stress under DR. However, linking abu genes that are known to be upregulated during the UPR and that acs-20 mutant is more sensitive to ER stress-inducing drug tunicamycin (Tm) is insufficient to make such claims. As ER stress induces the UPR pathway, the authors should characterize the activation of at least one branch of the UPR and/or conduct assays to demonstrate that the ER is under stress. There should be an accumulation of unfolded protein and/or aggregates. It is also possible that ER stress is induced by lipid perturbation at the ER as ACS-20 regulates lipids.

We thank Reviewer 1 for the insightful comments. Unlike the regular ER^{UPR} genes, the expression of which is significantly activated by transcription factors such as XBP-1 once triggered by various ER stressors, the *abu* genes were activated by the tunicamycin treatment only when the IRE-1 - XBP-1 ER^{UPR} pathway is blocked (Urano et al., 2002 DOI: 10.1083/jcb.200203086). Therefore, increased *abu* genes expression not only indicates animals are under ER stress, but also suggests organisms have problems dealing with the stress via the canonical ER^{UPR} mechanisms. We have tested this idea using the *hsp-4p::gfp* reporter, which is widely used to monitor the ER^{UPR} activation

especially for the XBP-1 branch. As shown in Fig. 3e, f, the *acs-20* mutation completely abolishes tunicamycin-induced activation of the *hsp-4p::gfp* reporter, suggesting mutant animals have compromised ER^{UPR} response.

We took advantages of the polyQ model developed by the Morimoto lab to assess the effect of *acs-20* mutation on unfolded protein. This model has been widely used in the field for this purpose. We quantitatively measured polyQ aggregates at the molecular level via filter trap assays (Fig. 4h, i), at the cellular level via imaging assays (Fig. 1d, e and Fig. 4 e, f), and at the animal level via paralysis assays (Fig. 1f and Fig. 4g). All experiments indicate that the *acs-20* mutation promotes polyQ aggregation. Therefore, these results strongly indicate that the *acs-20* mutation disrupts proteostasis, which leads to ER stress.

We have done a variety of experiments to examine the role of ACS-20 in lipid metabolism. Using lipid staining and imaging approaches, we demonstrate that the *acs-20* mutation has very little effect on triglyceride levels in the major lipid storage tissues (Supplementary Fig. 3a-d). Using biochemical approaches, we show that the *acs-20* mutation does not change the levels of most fatty acids but does affect certain phospholipids and sphingolipids (Supplementary Fig. 3e-g). One caveat of these biochemical assays is that whole animals were used to analyze the lipid profiles, whereas ACS-20's expression and functions are spatially restricted. Therefore, we currently cannot provide any direct evidence to connect the lipid metabolism and ER functions of ACS-20, but it is one of the future directions that we will pursue.

(2) Related to the major point #1, showing that tunicamycin delays development in acs-20 animals is not sufficient to link acs-20 to ER stress. As tunicamycin block N-glycosylation in the ER, many ER-resident proteins, proteins trafficked to other organelles, and secreted proteins will fail to fold in the ER, and they might be non-functional if they leave the ER. The ER of animals exposed to Tm will not be the only cell compartment affected but also other organelles part of the secretory pathway, the plasma membrane, and even the extracellular matrix will be negatively affected. Therefore, it cannot be excluded that the absence acs-20 together with Tm exacerbate the development of the animal independently of the ER state.

We agree with Reviewer 1 that the tunicamycin assays are not sufficient to link *acs-20* with ER stress as the drug functions through a specific mechanism that cannot rule out the alternative interpretation as the Reviewer proposed. Therefore, we have applied dithiothreitol and thapsigargin, two drugs that function through different mechanisms to cause ER stress, to various mutants and tested their effects on development. Like the tunicamycin treatment (Fig. 3g and Fig. 4d), both dithiothreitol and thapsigargin significantly delay

development when *acs-20* is mutated, and this effect can be reversed by the *ptr-8* mutation (Supplementary Fig. 4). Since experiments with three ER stressors show similar results, we concluded that ACS-20 functions through PTR-8 to affect animals' ER stress state.

*(3) The authors also link cytosolic proteostasis to PTR-8 and ACS-20. The data is insufficient to make such conclusion. The authors based their conclusion on (1) PTR-8 exacerbate Tm-induced development defect and (2) cytosolic polyQ accumulation is reduced in the absence of *acs-20* and *ptr-8* during DR. As described in major point #2, it is difficult to directly link Tm with proteostasis so the authors should be careful in their interpretation or to conduct more experiments. The difference in polyQ accumulation is interesting but there is a missing link of how the absence of both genes prevents the accumulation of aggregates.*

We have provided molecular evidence that the *acs-20* mutation leads to increased *ptr-8* expression (Fig. 4 and Fig. 5), and we have shown genetic evidence that PTR-8 mediates the effect of *acs-20* on proteostasis, ER stress and longevity (Fig. 5). PTR-8 is orthologous to Patched from the Hedgehog pathway, which has certain key components including downstream transcription factors missing but significant expansion of the Hedgehog receptors in *C. elegans*. Although the molecular connection between PTR-8 and proteostasis has not been elucidated in the current manuscript, it will help to provide a new angle to study functions of the non-canonical Hedgehog pathway.

*(4) The authors proposed a model where ACS-20 inhibits the expression of *ptr-8* via the transcription modulator NHR-23 and that in turns promote proteostasis and longevity. Based on their results, the links between each player are vague and I was anticipating getting more mechanist details from this study. Although their data convincingly demonstrate that the pathway is only relevant during DR, it is not clear to me how these two are link and what it the mechanism or cellular differences that will make their proposed pathway essential only during DR. Finally, it is also not clear why and how the role of ACS-20 during development is important to modulate lifespan. These different points should be addressed to increase the impact of the manuscript.*

We have revised the model in Fig. 6 based on both Reviewers' insightful suggestions. Besides the mechanisms downstream of PTR-8 that we have addressed in the previous major point, we agree with the Reviewer that there is a gap between ACS-20 and NHR-23 in the model. As mentioned in the Discussion, one appealing speculation is that ACS-20 might be involved in the biosynthesis of NHR-23's ligand through its function in lipid metabolism. RORA, the ortholog of NHR-23, uses cholesterol metabolic products as

ligands, whereas ACS-3, another long-chain acyl-CoA synthetase, directs the biosynthesis of phospholipid ligands for the NHR-25 nuclear receptor in epidermal cells. However, validation of this hypothesis requires detailed analysis of ACS-20's ACS enzymatic activities, and most likely forward genetic screen-based exploratory studies, which make it more suitable for an independent study.

As we mentioned in the Discussion, many key regulators of ageing function during development to affect lifespan and healthspan in adulthood. Therefore, it is not totally unexpected that ACS-20 functions during development to determine lifespan considering its expression is restricted mainly in the epidermis during development. In addition, the expression of NHR-23 also shows spatiotemporally restricted patterns. NHR-23 is expressed in the epidermis until the early L4 stage, after which it is expressed in the germline. Identification of the factors that shape the expression patterns of ACS-20 and NHR-23 will help to answer this important question from Reviewer 1.

Minor points to address

(1) I suggest the authors to report replicates in the reported bar chart. The number of samples and replicates should be indicated in figure legends.

We have revised the bar graphs to show replicates and reported sample sizes in figure legends. All the quantitative and statistical results have been included in the Supplementary Table 1.

Reviewer #2 (Remarks to the Author):

This manuscript describes a new mechanism by which dietary restriction (DR) extends life and health span in C. elegans. Specifically, loss of the lipid metabolism gene acs-20 prevents life span extension by various different ways of achieving DR, while expression of acs-20 in the hypodermic, one of its sites of natural expression, restores life span of the DR model eat-2. RNA-seq analysis revealed that proteostasis genes were induced in eat-2; acs-20 double mutants, and these mutants showed increased sensitivity to the tunicamycin, suggesting disturbed proteostasis. To identify mechanisms, the authors then tested deregulated genes in an RNAi screen and identified ptr-8, whose RNAi or mutation restored the extended life span of eat-2; acs-20, pinpointing increased ptr-8 expression as an inhibitor of life and health span. Finally, using another RNAi screen, the authors identify and validate by promoter GFP, qPCR, ChIP and promoter bashing the Nuclear Receptor NHR-23 as a negative regulator of ptr-8 expression, and show the deletion of a promoter element alone is sufficient to restore the long life span of eat-2; they also map ptr-8 action downstream of acs-20.

This is a very impressive manuscript and discovery overall. The mechanism is to my knowledge new and should be of great interest to scientists studying gene regulation, life span, and metabolism. The data are very well done - conclusions are mostly very well ascertained with several different DR regimens, RNAi and mutants, tissue specific RNAi and rescue, RNA-seq and qPCR, and both life span and health span, at least with a Q35 aggregation reporter and analysis, are analyzed. The experiments are mostly well controlled and analyzed, with a couple minor issues on controls and statistical comparisons.

We thank Reviewer 2 for the positive assessment of our study. We have carried out multiple experiments and revised the manuscript to address the Reviewer 2's concerns.

My only real concern revolves around the regulation of ptr-8. There seems to be some inconsistency with the description/rationale, and with the proposed mechanism.

1. Rationale: I'm confused how ptr-8 was found. The authors write on line 224 "The 95 genes that are significantly up-regulated in the eat-2; acs-20 vs. eat-2 comparison but not in the acs-20 vs. N2 comparison..." and later study these 95 genes by RNAi, finding ptr-8. However, Fig 4b clearly shows the ptr-8 doesn't belong in this category since it is clearly quite strongly and significantly induced in acs-20 vs. N2. Did the RNA-seq not show this regulatory pattern? This would be a concern, especially considering the very

strong and significant upregulation detected by qPCR in Fig 4d. Please show RNA-seq CPM or FPKM plots for ptr-8 for comparison with all 4 samples, with replicate data. The authors also later on line 301 state "significantly elevated ptr-8 expression upon nhr-23 or acs-20 RNAi...", so clearly acs-20 loss on its own causes ptr-8 upregulation. This is concerning as the authors' own data and argumentation appear incongruent. Or was there another reason why ptr-8 was studied?

We thank Reviewer 2 for bringing up this issue. Below is the FPKM plot for *ptr-8* from the mRNA-Seq data. It shows that *ptr-8* mRNA levels are elevated in *eat-2*; *acs-20* compared to *eat-2*, but in *acs-20* vs. N2. We believe the RT-qPCR results, which show *ptr-8* mRNA levels are increased by the *acs-20* mutation both under AL and DR conditions, with the *eat-2*; *acs-20* double mutant showing highest levels of *ptr-8* expression, are more reliable than the mRNA-Seq data as we have conducted the RT-qPCR experiments for many times with consistent results.

However, this discrepancy does not affect our conclusion that increased *ptr-8* expression is responsible for the lack of DR-induced lifespan extension in the *acs-20* mutant. Otherwise, we would expect either no difference or much reduced difference in lifespan of *ptr-8*; *acs-20* vs. *ptr-8 eat-2*; *acs-20*, which is contradictory to what we have reported in Fig. 4a.

2. Mechanism: I struggled to comprehend the proposed mechanism for NHR-23 action. Specifically, I don't understand how NHR-23 and the element in the ptr-8 promoter act on ptr-8 expression. Figure 5 shows that NHR-23 is required to repress ptr-8, as its RNAi induces the ptr-8 promoter::GFP reporter and endogenous ptr-8 mRNA, and hence depletion of NHR-23 shortens lifespan of eat-2 (wherein ACS-20 is active to repress ptr-8, dependent on nhr-23). So far so good. But then why does deletion of the element, through which NHR-23 is supposed to act, on its own not cause increased ptr-8 expression? If the element were deleted, then NHR-23 would not be able to

bind, and that should cause strong derepression, resembling nhr-23 loss. Why does this not happen? Along these lines, if NHR-23 represses a gene through an element in said gene's promoter, removal of both of these factors would not be expected to have opposite effects as is the case in Fig 5f. Perhaps their model is incomplete? An alternative model could be that NHR-23 prevents another TF, an activator, from acting through this element; this would also make sense of the ChIP data - if NHR-23 binding is reduced, then the other factor X might be more active, promoting increased ptr-8 expression and shorter life span. Such transcriptional repression of other TF's activities has been well described for the glucocorticoid receptor and NFkB, with GR exerting its anti-inflammatory effects by repressing NFkB's activation function. But a direct, repression-linked function for NHR-23 doesn't appear possible to me. I'd be very interested to hear the authors response to this comment.

We are very grateful for Reviewer 2's comments on NHR-23-mediated transcriptional regulation of *ptr-8*. After reviewing the data, we agree with Reviewer 2 that NHR-23 is unlikely to be the only and direct transcriptional regulator of *ptr-8*. Otherwise, deletion of the 1.9 - 2.7 kb *ptr-8* promoter region would lead to significantly elevated *ptr-8* expression and shortened lifespan. Therefore, we have revised the model in Results, Discussion, and legend for Fig. 6. The conclusions include: (1) NHR-23 is a transcriptional repressor of *ptr-8*; (2) the 1.9 - 2.7 kb *ptr-8* promoter region is a functional *cis*-regulatory element for its expression; (3) there might be a transcriptional activator of *ptr-8* that is negatively regulated by NHR-23.

Since *nhr-23* is a developmentally essential gene and double RNAi by mixing up two feeding RNAi *E. coli* strains yields inconsistent knockdown efficiency, we cannot do a relatively easy RNAi screen to identify this potential transcriptional activator of *ptr-8*. One feasible strategy might be using the *ptr-8* promoter::gfp reporter for an EMS-based forward genetic screen followed with whole genome sequencing to identify the target gene. However, it is beyond the scope of this work due to the time limit. Thus, we have decided to pursue this direction in future studies.

There are also some issues with missing controls where not all conditions have been assessed:

3. Fig 2h is missing controls; it would be ideal to include acs-20 and eat-2; acs-20 mutants. This would be important b/c it would inform on whether transgenic acs-20 expression can rescue acs-20 mutation; in fact, the authors should also show the experiment where they rescued acs-20 with a construct using its own promoter besides dpy-7; showing that acs-20 overexpression on its own promotes longevity would be relevant.

We thank Reviewer 2 for raising this point. When we performed the survival

assays shown in Fig. 2h, we had the *acs-20* and *eat-2*; *acs-20* samples as controls. We have updated the panel with the controls. We constructed the single-copy *acs-20* transgene driven by its own promoter for the rescue experiments. As shown in Supplementary Fig. 2, this transgene rescues the *acs-20* mutation both under AL and DR conditions. We did not observe any lifespan extension when *acs-20* is overexpressed by the single-copy transgene (Supplementary Fig. 2a). However, this is not conclusive regarding to the sufficiency of ACS-20 in lifespan regulation since we do not know whether low levels of ACS-20 overexpression will lead to enough gain-of-function to have an impact on lifespan.

4. Ideally, in Fig 4d, additional controls (ptr-8 single mutant and doubles with acs-20 and eat-2) would be added but I appreciate that this may not be possible after the fact.

We excluded those control groups to make the data presentation more focused in the initial submission. Now we have added them to the revised Fig. 4d with more detailed statistical analysis in Supplementary Table 1.

5. Similar comment for Fig 4e-g, it would be ideal to have additional controls (WT, eat-2, acs-20 singles with both control and ptr-8 RNAi), but this may not be possible in revision.

We have performed experiments to include all the control groups in the revised Fig. 4e-g.

Other

6. Fig 1c, e, 4b, etc for similar figures, please use appropriate ANOVA tests with multiple comparison corrections to assess all relevant comparisons, not just the indicated pairs.

We have performed ANOVA tests and reported the all the possible comparisons in Supplementary Table 1.

7. For RNA-seq data visualization, perhaps consider a Venn diagram, which might better highlight the 95 genes of interest. Also, why were downregulated genes not considered further?

We plotted a Venn diagram comparing genes upregulated in *eat-2*; *acs-20* vs. *eat-2* and genes not differentially expressed in *acs-20* vs. N2. We also made a list of the 95 genes of interest in Supplementary Table 2.

Genes that are downregulated in the *eat-2*; *acs-20* mutant compared to the *eat-2* mutant were not prioritized for functional studies because the most

conclusive experiments would be to construct transgenic overexpression lines in the *eat-2; acs-20* mutant to test which ones can extend lifespan. Since gene overexpression does not always lead to gain-of-function and the overexpression levels might require careful titration, we have decided to pursue this direction in future studies.

8. Methods, please add details and references on how GO term analysis was done.

We performed GO term analysis using the Metascape web-based service. We have added a paragraph to describe the details with a reference in the revised Methods part.

REVIEWER COMMENTS

Reviewer #1 (Remarks to the Author):

In this revised manuscript, the authors addressed my comments and other reviewers' comments adequately. They went to great lengths to address our comments experimentally as well as demonstrating the limitation of some approaches.

Reviewer #2 (Remarks to the Author):

The present version of the manuscript is improved and the authors have addressed most of my comments, including the revision of their mechanistic model, and the inclusion of new controls and statistical data analysis.

The key remaining is the expression analysis of ptr-8. The RNA-seq data in the rebuttal are clear: ptr-8 is not induced in acs-20 or eat-20 single mutants, only in the double mutant. RNA-seq is a highly reproducible and quantitative method and I thus place great value on these data. These RNA-seq data clearly and obviously contradict the qPCR data shown in Figure 4b, where both of the single mutants show upregulation vs. N2, with further upregulation in the double mutant.

Is this important? I think so. A central message of the manuscript is that high ptr-8 levels prevent lifespan extension downstream of acs-20 (and eat-2) mutation. However, the qPCR data make it clear that elevated levels of ptr-8 expression as seen in eat-2 are NOT a problem for longevity, as these worms live long. Perhaps this should be further addressed with an eat-2; ptr-8 mutant LSPN analysis - I assume here that this worm should live longer. Do the authors think there is a threshold of ptr-8 that cannot be exceeded? If so, why?

The explanation in the authors' rebuttal "We believe the RTqPCR results....are more reliable than the mRNA-Seq data as we have conducted the RT-qPCR experiments for many times with consistent results." is not satisfying - RNA-seq is a technically sophisticated method, and the authors used 3 replicates, which closely cluster, implying high quality data. The replicate number for qPCR "many times" is not specified in the manuscript. Indeed, this

should be stated clearly and for all experiments in the manuscript the authors should clearly indicated how many samples were analyzed (Figure legends). As well, the graph in Fig 4b should show individual data points. Ultimately, I remain concerned about the discrepancy of the RNA-seq and the qPCR data.

As well, it is clear from Fig 4A that the role of ptr-8 in reducing life span is not linked to dietary restriction, as the short life span of the acs-20 mutant is greatly alleviated by ptr-8 mutation. It is therefore clear that the role of ptr-8 is not solely in dietary restriction. Please revise conclusions accordingly (abstract, elsewhere).

Reviewer response summary

We thank both reviewers' comments on our manuscript. During the second round of revision, we have acquired new experimental evidence to strengthen the manuscript. Below we outlined our responses to individual reviewer's comments, which were italicized. We highlighted revisions in the manuscript with blue font.

Individual Reviewer Responses:

REVIEWER COMMENTS

Reviewer #1 (Remarks to the Author):

In this revised manuscript, the authors addressed my comments and other reviewers' comments adequately. They went to great lengths to address our comments experimentally as well as demonstrating the limitation of some approaches.

We thank Reviewer 1 for the positive assessment of our study.

Reviewer #2 (Remarks to the Author):

The present version of the manuscript is improved and the authors have addressed most of my comments, including the revision of their mechanistic model, and the inclusion of new controls and statistical data analysis.

The key remaining is the expression analysis of ptr-8. The RNA-seq data in the rebuttal are clear: ptr-8 is not induced in acs-20 or eat-20 single mutants, only in the double mutant. RNA-seq is a highly reproducible and quantitative method and I thus place great value on these data. These RNA-seq data clearly and obviously contradict the qPCR data shown in Figure 4b, where both of the single mutants show upregulation vs. N2, with further upregulation in the double mutant.

Is this important? I think so. A central message of the manuscript is that high ptr-8 levels prevent lifespan extension downstream of acs-20 (and eat-2) mutation. However, the qPCR data make it clear that elevated levels of ptr-8 expression as seen in eat-2 are NOT a problem for longevity, as these worms live long. Perhaps this should be further addressed with an eat-2; ptr-8 mutant LSPN analysis - I assume here that this worm should live longer. Do the authors think there is a threshold of ptr-8 that cannot be exceeded? If so, why?

We thank Reviewer 2 for the insightful comments on the discrepancy between

mRNA-Seq data and qPCR validation. We have carefully examined the mRNA-Seq results, and we found the expression levels of *pmp-2*, which we used as the internal control for qPCR normalization, show variability among different genetic backgrounds. We then performed three independent RT-qPCR experiments to measure *ptr-8* mRNA levels with *act-1*, another housekeeping gene that has been widely used as the internal control in the field, for data normalization. As shown in the revised Fig 4b, *ptr-8* mRNA levels are significantly elevated (nearly 10-fold) in the *eat-2; acs-20* double mutant compared to *eat-2*; whereas the *acs-20* mutant has slightly increased (less than 50%) *ptr-8* expression compared to the wild-type N2. These RT-qPCR results are consistent with the mRNA-Seq data that *ptr-8* expression is activated by the *acs-20* mutant under DR. We thank Reviewer 2 again for being persistent on this point, which helps us to strengthen the manuscript.

The explanation in the authors' rebuttal "We believe the RTqPCR results....are more reliable than the mRNA-Seq data as we have conducted the RT-qPCR experiments for many times with consistent results." is not satisfying - RNA-seq is a technically sophisticated method, and the authors used 3 replicates, which closely cluster, implying high quality data. The replicate number for qPCR "many times" is not specified in the manuscript. Indeed, this should be stated clearly and for all experiments in the manuscript the authors should clearly indicated how many samples were analyzed (Figure legends). As well, the graph in Fig 4b should show individual data points. Ultimately, I remain concerned about the discrepancy of the RNA-seq and the qPCR data.

We apologize for using ambiguous language to describe the experiments in the previous rebuttal letter. We have added the replicate numbers in the revised figure legends.

As well, it is clear from Fig 4A that the role of ptr-8 in reducing life span is not linked to dietary restriction, as the short life span of the acs-20 mutant is greatly alleviated by ptr-8 mutation. It is therefore clear that the role of ptr-8 is not solely in dietary restriction. Please revise conclusions accordingly (abstract, elsewhere).

We agree with Reviewer 2 that the *ptr-8* mutation increases lifespan of the *acs-20* mutant. However, lifespan of the *eat-2; acs-20* double mutant is increased by the *ptr-8* mutation to a higher level, indicating that the *ptr-8* mutation restores the lifespan extension effect of DR in the *acs-20* mutant. Therefore, in the absence of ACS-20, PTR-8 plays a specific role in DR-induced longevity rather than lifespan regulation in general. We have revised the manuscript to make the statement more clear.

REVIEWERS' COMMENTS

Reviewer #2 (Remarks to the Author):

The authors have addressed all criticisms to my satisfaction.